# Precise Diffusion Inversion: Towards Novel Samples and Few-Step Models

**Jing Zuo   Luoping Cui   Chuang Zhu   Yonggang Qi** ✉

School of Artificial Intelligence, Beijing University of Posts and Telecommunications

zuoj0723@gmail.com, {lpcui,czhu,qiyg}@bupt.edu.cn

✉ Corresponding author

## Abstract

The diffusion inversion problem seeks to recover the latent generative trajectory of a diffusion model given a real image. Faithful inversion is critical for ensuring consistency in diffusion-based image editing. Prior works formulate this task as a fixed-point problem and solve it using numerical methods. However, achieving both accuracy and efficiency remains challenging, especially for few-step models and novel samples. In this paper, we propose *PreciseInv*, a general-purpose test-time optimization framework that enables fast and faithful inversion in as few as two inference steps. Unlike root-finding methods, we reformulate inversion as a learning problem and introduce a dynamic programming-inspired strategy to recursively estimate a parameterized sequence of noise embeddings. This design leverages the smoothness of the diffusion latent space for accurate gradient-based optimization and ensures memory efficiency via recursive subproblem construction. We further provide a theoretical analysis of *PreciseInv*'s convergence and derive a provable upper bound on its reconstruction error. Extensive experiments on COCO 2017, DarkFace, and a stylized cartoon dataset show that *PreciseInv* achieves state-of-the-art performance in both reconstruction quality and inference speed. Improvements are especially notable for few-step models and under distribution shifts. Moreover, precise inversion yields substantial gains in editing consistency for text-driven image manipulation tasks. Code is available at `https://github.com/panda7777777/PreciseInv`

## 1   Introduction

Large-scale pre-trained diffusion models [10, 17, 32, 33] have demonstrated strong generative capabilities in producing high-quality and diverse images. Built upon these models, many recent methods enable a wide range of image editing operations by guiding the generative process with user inputs, such as text prompts [4, 9, 16, 21, 22, 29], reference images [1, 6, 23, 25], and spatial masks [8, 28]. To ensure consistency and controllability in such edits, faithful inversion of real images into the diffusion process is essential [13].

Since the sampling process of diffusion models is inherently irreversible, mapping a real image back into the model domain remains a fundamental challenge. Existing methods on diffusion inversion can be divided into three major categories: forward optimization, backward optimization, and invertible samplers. Invertible samplers, such as EDICT [38], BDIA [43], and BELM [39], introduce explicit constraints into the sampling procedure to enable bidirectional generation. Backward optimization methods [21, 22, 29] fix the forward noising trajectory of a real image and optimize the reverse diffusion process to match this trajectory. Both types of methods improve reconstruction quality by modifying the denoising path, but often weaken the generative ability of the diffusion model. To address this issue, forward optimization methods estimate a noising trajectory that can be faithfully

39th Conference on Neural Information Processing Systems (NeurIPS 2025).

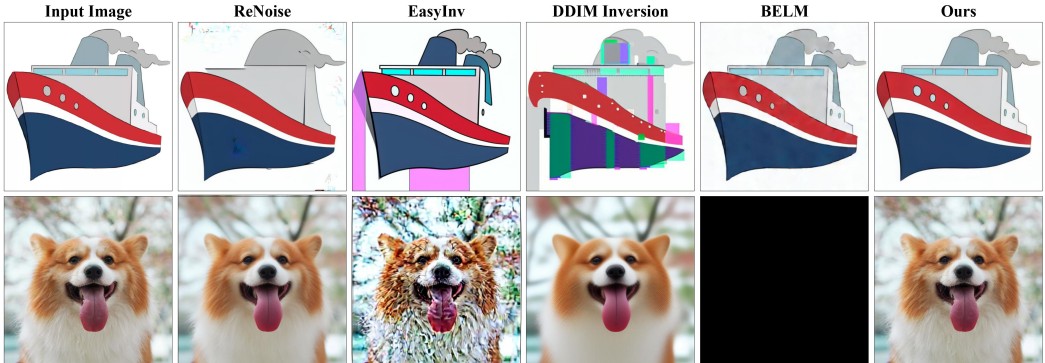

| Input Image | ReNoise | EasyInv | DDIM Inversion | BELM | Ours |

Figure 1: Qualitative comparison of diffusion inversion methods using SD v1.4 on challenging cases: novel sample (first row) and two-step inference (second row).

reversed by the diffusion model. For example, several methods [30, 34, 45] convert the problem into a fixed-point equation and solve it via Anderson acceleration [2], while ReNoise [13] refines an iterative renoising mechanism. However, these methods suffer from slow convergence and limited performance, especially under few-step inference or on novel samples, as illustrated in Fig. 1.

In this paper, we propose a general framework named *PreciseInv* that efficiently solves the diffusion inversion problem by progressively learning a parameterized noising trajectory. Specifically, we reformulate diffusion inversion as a learning problem and decompose it into $T$ overlapping subproblems by leveraging the Markov property of the diffusion process. We then introduce a dynamic programming-inspired strategy [5] to recursively solve these subproblems. In each subproblem, *PreciseInv* learns a local noise embedding that approximates the optimal intermediate state along the inversion trajectory. Benefiting from the smooth transitions in the latent space [44], this gradient-based optimization converges more accurately than numerical solvers. Moreover, the recursive formulation is highly memory-efficient: inversion on the SD v1.4 base model requires only 3.43 GB of GPU memory, regardless of the number of inference steps. Extensive experiments show that *PreciseInv* achieves state-of-the-art reconstruction performance in both quality and efficiency (Section 4.1). The improvements are especially pronounced under few-step inference (e.g., $T{=}2, 4$) and on novel samples. Furthermore, precise inversion yields more semantically continuous latent representations (Fig. 5), which substantially improve consistency and controllability in text-driven editing tasks (Section 4.2).

Our main contributions are summarized as follows: (i) We propose *PreciseInv*, a general-purpose test-time optimization framework for diffusion inversion that enables fast and faithful image reconstruction under few-step inference and distribution shift. (ii) We reformalize diffusion inversion as a sequence of non-overlapping learning subproblems and introduce a dynamic programming strategy to solve them efficiently. (iii) We demonstrate that precise inversion yields smoother latent trajectories, which in turn improve consistency and controllability in text-driven image editing.

## 2 Background

### 2.1 Diffusion Models

Given samples from the data distribution $q(\mathbf{x}_0)$, diffusion models [17] define a generative process by learning to reverse a fixed Markovian forward process that progressively corrupts data by adding Gaussian noise. The forward process is formulated as:

$$q(\mathbf{x}_{1:T} \mid \mathbf{x}_0) := \prod_{t=1}^{T} q(\mathbf{x}_t \mid \mathbf{x}_{t-1}), \quad q(\mathbf{x}_t \mid \mathbf{x}_{t-1}) := \mathcal{N}(\mathbf{x}_t; \sqrt{\alpha_t}\, \mathbf{x}_{t-1}, (1 - \alpha_t)\mathbf{I}), \qquad (1)$$

where $\mathbf{x}_0 \sim q(\mathbf{x}_0)$, $\{\mathbf{x}_t\}_{t=1}^{T}$ are latent variables, and $\{\alpha_t\}_{t=1}^{T} \in (0, 1]^T$ is a predefined noise schedule. The marginal distribution at an arbitrary timestep $t$ is given by:

$$q(\mathbf{x}_t \mid \mathbf{x}_0) = \mathcal{N}(\mathbf{x}_t; \sqrt{\bar{\alpha}_t}\, \mathbf{x}_0, (1 - \bar{\alpha}_t)\mathbf{I}), \qquad (2)$$

where $\bar{\alpha}_t := \prod_{s=1}^{t} \alpha_s$. The generative process is modeled as a parameterized Markov chain:

$$p_\theta(\mathbf{x}_{0:T}) := p(\mathbf{x}_T) \prod_{t=1}^{T} p_\theta(\mathbf{x}_{t-1} \mid \mathbf{x}_t), \quad p_\theta(\mathbf{x}_{t-1} \mid \mathbf{x}_t) := \mathcal{N}(\mathbf{x}_{t-1}; \boldsymbol{\mu}_\theta(\mathbf{x}_t, t), \boldsymbol{\Sigma}_\theta(\mathbf{x}_t, t)), \quad (3)$$

where $p(\mathbf{x}_T) = \mathcal{N}(\mathbf{x}_T; \mathbf{0}, \mathbf{I})$ and $\theta$ denotes the model parameters. In practice, a denoising neural network $\boldsymbol{\epsilon}_\theta$ is trained to predict $\boldsymbol{\mu}_\theta(\mathbf{x}_t, t)$, and $\boldsymbol{\Sigma}_\theta(\mathbf{x}_t, t)$ is set to a constant $\sigma_t$ according to $\alpha_t$. The training objective minimizes a simplified variational bound:

$$\mathcal{L}_{\text{simple}}(\theta) := \mathbb{E}_{t \sim \mathcal{U}\{1,\dots,T\}, \mathbf{x}_0 \sim q(\mathbf{x}_0), \boldsymbol{\epsilon} \sim \mathcal{N}(\mathbf{0}, \mathbf{I})} \left[ \|\boldsymbol{\epsilon} - \boldsymbol{\epsilon}_\theta(\mathbf{x}_t, t)\|^2 \right], \quad (4)$$

where

$$\mathbf{x}_t = \sqrt{\bar{\alpha}_t} \, \mathbf{x}_0 + \sqrt{1 - \bar{\alpha}_t} \, \boldsymbol{\epsilon}. \quad (5)$$

Starting from $\mathbf{x}_T \sim \mathcal{N}(\mathbf{x}_T; \mathbf{0}, \mathbf{I})$, the DDPM sampling procedure [17] iteratively applies the reverse step to generate $\mathbf{x}_0$ as follows:

$$\mathbf{x}_{t-1} = \frac{1}{\sqrt{\alpha_t}} \left( \mathbf{x}_t - \frac{\sqrt{1 - \alpha_t}}{\sqrt{1 - \alpha_{t-1}}} \boldsymbol{\epsilon}_\theta(\mathbf{x}_t, t) \right) + \sigma_t \mathbf{z}, \quad (6)$$

where $\mathbf{z} \sim \mathcal{N}(\mathbf{z}; \mathbf{0}, \mathbf{I})$. To reduce the number of inference steps, the DDIM sampler [36] converts Eq. (6) into a non-Markovian form:

$$\mathbf{x}_{t-1} = \sqrt{\frac{\alpha_{t-1}}{\alpha_t}} \mathbf{x_t} + \left( \sqrt{\frac{1}{\alpha_{t-1}} - 1} - \sqrt{\frac{1}{\alpha_t} - 1} \right) \boldsymbol{\epsilon}_\theta(\mathbf{x}_t, t). \quad (7)$$

To further enhance the inference efficiency, recent efforts have focused on training few-step diffusion models, such as Consistency Models (CM) [37], Latent Consistency Models (LCM) [27], Adversarial Diffusion Distillation (ADD) models [35], Shortcut models [12] and MeanFlow [14].

## 2.2 Image Inversion in Diffusion Models

Inverting real images into the latent space of a pretrained diffusion model is critical for diffusion-based image editing methods. Given a real image $\mathbf{x}_0$, our goal is to estimate a corresponding noisy latent $\mathbf{x}_T$, such that $\mathbf{x}_0$ can be reconstructed by applying the reversed denoising trajectory to $\mathbf{x}_T$. Unfortunately, the neural network $\boldsymbol{\epsilon}_\theta$ do not inherently predict noise direction $\boldsymbol{\epsilon}$ from $\mathbf{x}_{t-1}$ to $\mathbf{x}_t$. DDIM Inversion [8] was among the earliest methods attempted to address this issue. Rewriting the Eq. (7), we obtain:

$$\mathbf{x}_t = \sqrt{\frac{\alpha_t}{\alpha_{t-1}}} \mathbf{x}_{t-1} - \sqrt{\frac{\alpha_t}{\alpha_{t-1}}} \left( \sqrt{\frac{1}{\alpha_{t-1}} - 1} - \sqrt{\frac{1}{\alpha_t} - 1} \right) \boldsymbol{\epsilon}_\theta(\mathbf{x}_t, t). \quad (8)$$

DDIM Inversion assumes $\boldsymbol{\epsilon}_\theta(\mathbf{x}_{t-1}, t-1) \approx \boldsymbol{\epsilon}_\theta(\mathbf{x}_t, t)$ and iteratively applies Eq. (8) from $t = 1$ to $t = T$. However, the denoising trajectory is nonlinear, resulting in discrepancies between predicted $\hat{\mathbf{x}}_t$ and actual $\mathbf{x}_t$.

Subsequently, there are three main lines of work on diffusion inversion: modifying the reversed denoising trajectory, designing invertible samplers or optimizing the initial noise latent $\mathbf{x}_T$. A common practice of the former is to optimize the denoising process to better align with the trajectory obtained by DDIM inversion. For example, Null-Text Inversion [29] optimizes unconditional text embeddings to implicitly adjust the denoising trajectory, while Eta Inversion [22] uses a hyperparameter $\eta$ to control the trade-off between the stochastic denoising trajectory and the vanilla DDIM Inversion trajectory. Moreover, a DDPM inversion method [21] was proposed, which rectifies the DDPM denoising process using a forward trajectory constructed by independently applying Eq.(5) at each timestep $t$. More recently, several invertible samplers [38, 39, 43] have been proposed for bidirectional generation. While, in general, manually modifying the denoising trajectory enhancing reconstruction quality, it inevitably disrupts the original generative process of the diffusion model.

To preserve the original generative process, another line of methods has been proposed to estimate a more accurate value of $\boldsymbol{\epsilon}_\theta(\mathbf{x}_t, t)$ in Eq. (8). For example, ReNoise [13] ensembles multiple DDIM Inversion results at each iteration. In addition, several methods reformulates the diffusion inversion

problem as a fixed-point problem and solves the problem by improving Anderson acceleration [2] methods. Let the estimated value of $\epsilon_\theta(\mathbf{x}_t, t)$ be denoted as $\epsilon_t^*$, then we have:

$$\epsilon_t^* = \epsilon_\theta(\mathbf{x}_t, t). \tag{9}$$

Substituting Eq. (9) into Eq. (8), we obtain:

$$\mathbf{x}_t = \sqrt{\frac{\alpha_t}{\alpha_{t-1}}} \mathbf{x}_{t-1} - \sqrt{\frac{\alpha_t}{\alpha_{t-1}}} \left( \sqrt{\frac{1}{\alpha_{t-1}} - 1} - \sqrt{\frac{1}{\alpha_t} - 1} \right) \epsilon_t^*. \tag{10}$$

By replacing $\mathbf{x}_t$ in Eq. (9) with Eq. (10), we deduce that:

$$\epsilon_t^* = \epsilon_\theta \left( \sqrt{\frac{\alpha_t}{\alpha_{t-1}}} \mathbf{x}_{t-1} - \sqrt{\frac{\alpha_t}{\alpha_{t-1}}} \left( \sqrt{\frac{1}{\alpha_{t-1}} - 1} - \sqrt{\frac{1}{\alpha_t} - 1} \right) \epsilon_t^*, t \right). \tag{11}$$

To effectively solve Fixed-Point Eq.(11), AIDI[30] uniformly mixes the outputs from two adjacent iterations in vanilla Anderson acceleration, while EasyInv [45] increases the influence of $\mathbf{x}_0$ in estimating $\mathbf{x}_T^*$ by directly predicting $\mathbf{x}_t^*$ instead of $\epsilon_t^*$. Despite recent progress, achieving precise and efficient inversion remains challenging, especially under few-step inference or when the inputs are novel, as illustrated in Fig. 1.

## 3 Method

### 3.1 Reformulating Diffusion Inversion as a Learning Problem

Given a pretrained diffusion model that defines a mapping $\mathcal{X} : \mathbf{x}_T \mapsto \mathbf{x}_{T-1} \mapsto \cdots \mapsto \mathbf{x}_0$ from latent space to image space, we hypothesize that the inverse mapping $\mathcal{X}^{-1} : \mathbf{x}_0 \mapsto \mathbf{x}_1 \mapsto \cdots \mapsto \mathbf{x}_T$ exhibits a smooth optimization landscape, owing to the smoothness of transitions in the latent space [44]. Based on this observation, we reformulate diffusion inversion as a learning problem. Let $\mathbf{x}_0$ be an observed image, and let $\epsilon_T^*$ denote a learnable noise embedding. The objective is defined as:

$$\arg \min_{\epsilon_T^*} \| \mathcal{X}(\mathbf{x}_T^*) - \mathbf{x}_0 \|^2, \tag{12}$$

where

$$\mathbf{x}_T^* = \sqrt{\bar{\alpha}_T} \, \mathbf{x}_0 + \sqrt{1 - \bar{\alpha}_T} \, \epsilon_T^*. \tag{13}$$

### 3.2 Progressive Denoising Trajectory Learning

Directly optimizing (12) requires backpropagation through all $T$ timesteps, which incurs prohibitive memory and computational costs. To address this, we exploit the Markov property of the diffusion process to decompose problem (12) into $T$ subproblems and solve them efficiently using dynamic programming principles.

**Subproblem Decomposition** Let $\{\mathbf{x}_t\}_{t=1}^T$ denote latent variables along the denoising trajectory from $\mathbf{x}_0$ to $\mathbf{x}_T$, as indicated by the spherical markers along the blue arrow in Fig. 2. Since only $\mathbf{x}_0$ is observed, we adopt a bottom-up dynamic programming strategy to sequentially estimate latent states from $\mathbf{x}_1$ to $\mathbf{x}_T$. To this end, we define a collection of subproblems $\{\mathcal{P}(t)\}_{t=1}^T$, where each $\mathcal{P}(t)$ corresponds to the estimation of $\mathbf{x}_t$. Specifically,

$$\mathcal{P}(t) = \arg \min_{\epsilon_t^*} \| \mathcal{X}(\mathbf{x}_t^*) - \mathbf{x}_0 \|^2, \tag{14}$$

with

$$\mathbf{x}_t^* = \sqrt{\bar{\alpha}_t} \, \mathbf{x}_0 + \sqrt{1 - \bar{\alpha}_t} \, \epsilon_t^*, \tag{15}$$

where $\epsilon_t^*$ is the learnable noise embedding at timestep $t \in \{1, 2, \ldots, T\}$.

**Optimal Substructure** The inverse mapping $\mathcal{X}^{-1} : \mathbf{x}_0 \mapsto \mathbf{x}_1 \mapsto \cdots \mapsto \mathbf{x}_T$ inherits the Markov structure of $\mathcal{X}$, implying that the optimal solution to each subproblem $\mathcal{P}(t)$ depends only on the solution to the previous subproblem $\mathcal{P}(t-1)$ for all $t \geq 2$. In particular, $\mathcal{P}(1)$ depends solely on the known observation $\mathbf{x}_0$. This recursive dependency establishes that the subproblems $\{\mathcal{P}(t)\}_{t=1}^T$ exhibit the optimal substructure property.

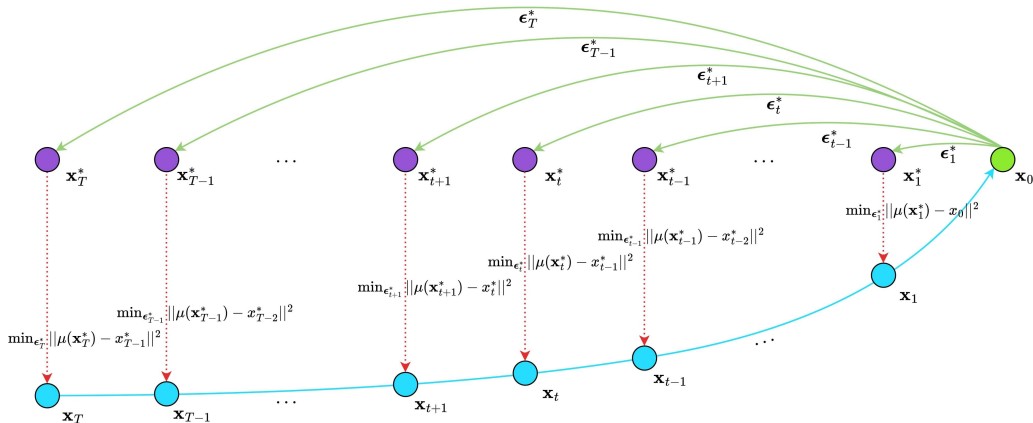

Figure 2: Overview of the proposed *PreciseInv*.

**Overlapping Subproblems** Each subproblem $\mathcal{P}(t)$ seeks a latent variable $\mathbf{x}_t$ such that the denoising trajectory from $\mathbf{x}_t$ accurately reconstructs $\mathbf{x}_0$. As adjacent subproblems $\mathcal{P}(t)$ and $\mathcal{P}(t-1)$ share this reconstruction objective and lie on successive steps of the trajectory, their optimization processes overlap. Specifically, the solution to $\mathcal{P}(t-1)$ influences and partially informs the optimization of $\mathcal{P}(t)$, resulting in substantial overlap across subproblems.

**Bottom-Up Computation** Without loss of generality, we adopt the DDIM sampling strategy. For convenience in the subsequent analysis, we reframe Eq. (7) as:

$$\mu(\mathbf{x}_t) = \sqrt{\frac{\alpha_{t-1}}{\alpha_t}}\mathbf{x}_t + \left(\sqrt{\frac{1}{\alpha_{t-1}} - 1} - \sqrt{\frac{1}{\alpha_t} - 1}\right)\boldsymbol{\epsilon}_\theta(\mathbf{x}_t, t). \tag{16}$$

Substituting into Eq. (14), the first subproblem becomes:

$$\begin{aligned}
\mathcal{P}(1) &= \arg\min_{\boldsymbol{\epsilon}_1^*} \|\mathcal{X}(\mathbf{x}_1^*) - \mathbf{x}_0\|^2 \\
&= \arg\min_{\boldsymbol{\epsilon}_1^*} \|\mu(\mathbf{x}_1^*) - \mathbf{x}_0\|^2.
\end{aligned} \tag{17}$$

Exploiting the optimal substructure of the subproblems $\{\mathcal{P}(t)\}_{t=1}^T$, we derive a recursive formulation $f$ that maps $\mathcal{P}(t-1)$ to $\mathcal{P}(t)$:

$$\begin{aligned}
\mathcal{P}(t) &= \arg\min_{\boldsymbol{\epsilon}_t^*} \|\mathcal{X}(\mathbf{x}_t^*) - \mathbf{x}_0\|^2 \\
&= \arg\min_{\boldsymbol{\epsilon}_t^*} \|\mu(\mathbf{x}_t^*) - \mathbf{x}_{t-1}^*\|^2 \\
&= f(\mathcal{P}(t-1)),
\end{aligned} \tag{18}$$

where $\mathbf{x}_{t-1}^*$ denotes the optimized solution obtained from $\mathcal{P}(t-1)$. Leveraging the recursive formulation $f$, we solve the subproblems $\{\mathcal{P}(t)\}_{t=1}^T$ iteratively from $t = 1$ to $t = T$, as indicated by the red dotted arrows in Fig. 2. We define a convergence threshold $\eta$ for $\mathcal{P}(t)$, and stop the optimization when $\|\mu(\mathbf{x}_t^*) - \mathbf{x}_{t-1}^*\|^2 < \eta$. In practice, we set $\eta$ in the range of $10^{-2}$ to $10^{-5}$. $\eta$ and the inference step number $T$ jointly control the trade-off between reconstruction accuracy and inference speed, as detailed in Table 5.

**Optimization Procedure.** The overall optimization algorithm of *PreciseInv* is presented in Algorithm 1. The procedure is memory-efficient since backpropagation is performed only over a single timestep. Moreover, *PreciseInv* is general and model-agnostic. We show its adaptability to different diffusion backbones (e.g., rectified flow [26]) and samplers (e.g., DDPM [17]) in Appendix A, and provide additional quantitative results in Appendix D.1.

**Algorithm 1** *PreciseInv* for Diffusion Models with DDIM Sampler
___
1: **Input:** Real image $\mathbf{x}_0$, diffusion model $\epsilon_\theta$, convergence threshold $\eta$, number of inference steps $T$
2: **Output:** Inverted latent $\mathbf{x}_T^*$
3: **for** $t = 1$ to $T$ **do**
4:      Initialize $\epsilon_t^* \sim \mathcal{N}(\mathbf{0}, \mathbf{I})$
5:      **if** $t > 1$ **then**
6:          $\mathbf{x}_{t-1}^* \leftarrow \sqrt{\bar{\alpha}_{t-1}}\,\mathbf{x}_0 + \sqrt{1 - \bar{\alpha}_{t-1}}\,\epsilon_{t-1}^*$
7:      **else**
8:          $\mathbf{x}_{t-1}^* \leftarrow \mathbf{x}_0$
9:      **end if**
10:     // Apply a single DDIM Inversion step
11:     $\epsilon_t^* \leftarrow \frac{1}{\sqrt{1-\bar{\alpha}_t}}\left(\mu(\mathbf{x}_{t-1}^*) - \sqrt{\bar{\alpha}_t}\mathbf{x}_0\right)$     ($\mu$ defined in Eq. (16))
12:     $\mathbf{x}_t^* \leftarrow \sqrt{\bar{\alpha}_t}\,\mathbf{x}_0 + \sqrt{1 - \bar{\alpha}_t}\,\epsilon_t^*$
13:     $\mathcal{L}_{\text{rec}} \leftarrow \|\mu(\mathbf{x}_t^*) - \mathbf{x}_{t-1}^*\|^2$
14:     **while** $\mathcal{L}_{\text{rec}} < \eta$ **do**
15:         $\mathbf{x}_t^* \leftarrow \sqrt{\bar{\alpha}_t}\,\mathbf{x}_0 + \sqrt{1 - \bar{\alpha}_t}\,\epsilon_t^*$
16:         $\mathcal{L}_{\text{rec}} \leftarrow \|\mu(\mathbf{x}_t^*) - \mathbf{x}_{t-1}^*\|^2$
17:         $\epsilon_t^* \leftarrow \epsilon_t^* - \nabla\mathcal{L}_{\text{rec}}$
18:     **end while**
19: **end for**
20: **return** $\mathbf{x}_T^*$
___

## 3.3 Theoretical Analysis of Convergence

While the progressive optimization strategy above is conceptually appealing, its convergence has yet to be formally established. Here, we provide a theoretical proof to establish its convergence. We begin by stating two standard assumptions that are widely used in diffusion models [37].

**Assumption 1.** *The diffusion model $\epsilon_\theta(\mathbf{x}_t, t)$ is Lipschitz continuous in $\mathbf{x}_t$ with constant $L_m$, i.e., $\|\epsilon_\theta(\mathbf{x}_1, t) - \epsilon_\theta(\mathbf{x}_2, t)\| \leq L_m \|\mathbf{x}_1 - \mathbf{x}_2\|$.*

**Assumption 2.** *Let $\mathcal{L}_t := \|\mu(\mathbf{x}_t) - \mathbf{x}_{t-1}\|^2$. The function $\mathcal{L}_t$ has L-Lipschitz continuous gradients with respect to $\mathbf{x}_t$, i.e., $\|\nabla\mathcal{L}_t(\mathbf{x}_t) - \nabla\mathcal{L}_t(\mathbf{x}_t')\| \leq L \|\mathbf{x}_t - \mathbf{x}_t'\|$.*

Under these assumptions, standard non-convex optimization results [15] apply. With gradient descent using a fixed step size $\gamma < 2/L$ for each subproblem $\mathcal{P}(t)$, the local loss $\mathcal{L}_t$ decreases monotonically. After $K$ iterations,

$$\min_{k \in [1, K]} \|\nabla\mathcal{L}_t^{(k)}\|^2 \leq \mathcal{O}\left(\tfrac{1}{K}\right), \tag{19}$$

indicating convergence to a stationary point $\delta_t$. Since each step minimizes a local reconstruction loss and the optimization proceeds recursively, the global loss $\mathcal{L}$ is non-increasing. Given that $\mu$ is Lipschitz in $\mathbf{x}_t$, the accumulated reconstruction error satisfies:

$$\|\mathcal{X}(\mathbf{x}_t) - \mathbf{x}_0\|^2 \leq \sum_{t=1}^{T}\left(\prod_{j=1}^{t-1} L_m^{(j)}\right)\delta_t. \tag{20}$$

If $L_m^{(j)} \leq L_m < 1$ (satisfies due to Assumption 1) and let $\delta = \max\{\delta_1, \cdots, \delta_T\}$, a geometric upper bound follows:

$$\|\mathcal{X}(\mathbf{x}_T) - \mathbf{x}_0\|^2 \leq \delta \sum_{t=1}^{T} L_m^{t-1} \leq \tfrac{\delta}{1 - L_m}. \tag{21}$$

**Theorem 1.** *Under Assumptions 1 and 2, gradient descent on $\mathcal{P}(t)$ with step size $\gamma < 2/L$ converges monotonically to a stationary point of $\mathcal{L}_t$.*

**Theorem 2.** *Let $\Gamma(t) := \|\mathcal{X}(\mathbf{x}_t) - \mathbf{x}_0\|^2$. Under Assumption 1 and Theorem 1, $\Gamma(T)$ admits a geometric upper bound, i.e., $\Gamma(T) \leq \delta/(1 - L_m)$.*

Detailed proofs of Theorem 1 and 2 are provided in Appendix B.

### 3.4 Prompt-driven Image Editing

We apply *PreciseInv* to enable prompt-driven editing by replacing the inversion stage without modifying the editing pipeline. Let $\mathbf{x}_0$ denote the input image, and let $c_s$ and $c_t$ denote the source and target prompts, respectively. We perform inversion conditioned on $c_s$, following the progressive optimization strategy in Eq. (14). At each step $t$, we solve

$$\mathcal{P}_{\text{edit}}(t) = \arg\min_{\boldsymbol{\epsilon}_t^*} \left\| \mu_{c_s}(\mathbf{x}_t^*) - \mathbf{x}_{t-1}^* \right\|^2, \tag{22}$$

where $\mu_c(\cdot)$ generalizes Eq. (16) to conditional generation. This yields a latent $\mathbf{x}_T^*$ that faithfully reconstructs $\mathbf{x}_0$ under the conditioning of the source prompt.

To generate the edited image, we reuse $\mathbf{x}_T^*$ and apply DDIM sampling under classifier-free guidance (CFG) [18], where the target and source prompts act as positive and negative conditioning signals, respectively. The conditional mean of the guided denoising process is:

$$\mu_{c_{\text{pos}}, c_{\text{neg}}}(\mathbf{x}_t) = \sqrt{\frac{\alpha_{t-1}}{\alpha_t}}\,\mathbf{x}_t + \left( \sqrt{\frac{1}{\alpha_{t-1}} - 1} - \sqrt{\frac{1}{\alpha_t} - 1} \right) \left[ (1+\omega)\,\boldsymbol{\epsilon}_\theta(\mathbf{x}_t, t, c_{\text{pos}}) - \omega\,\boldsymbol{\epsilon}_\theta(\mathbf{x}_t, t, c_{\text{neg}}) \right],$$

$$\tag{23}$$

where $\mathcal{X}_{c_{\text{pos}}, c_{\text{neg}}}$ denotes the deterministic denoising trajectory governed by Eq. (23). This pipeline procedure enables prompt-driven editing without task-specific designs and improves fidelity and localization by aligning the initial latent more precisely with the structure of the input image.

## 4 Experiments

In this section, we present experimental results with the aim of (i) demonstrating the state-of-the-art performance of *PreciseInv* for image reconstruction in terms of both quality and efficiency; (ii) verifying that faithful inversion enhances consistency in prompt-guided image editing; and (iii) providing insights into its local behaviors and interaction mechanisms.

### 4.1 Image Reconstruction

**Datasets.**   We evaluate *PreciseInv* on three domains: LAION-aligned, low-light, and stylized. (i) The COCO 2017 [24] validation set contains 5,000 natural images with diverse everyday scenes. It serves as a close proxy to the LAION distribution used in training most text-to-image diffusion models. (ii) The DarkFace [41] validation set includes 6,089 nighttime images captured in real-world low-light conditions, exhibiting extreme visibility degradation. (iii) The Cartoon dataset consists of 722 stylized images collected from the internet. These samples exhibit abstract shapes, exaggerated structures, and vivid palettes, reflecting rare visual styles far from the training distribution.

**Baselines.**   We compare our method with forward optimization methods, including *DDIM Inversion* [8], *ReNoise* [13], and *EasyInv* [45]; and invertible samplers, including *EDICT* [38], *BDIA* [43], and *BELM* [39]. Following prior work [45, 39], we don't compare with backward optimization methods, as they are not designed for recovering the original generative trajectory of diffusion models. We carefully tune hyperparameters for all methods.

**Metrics.**   We evaluate reconstruction quality using LPIPS, SSIM, and RSNR, which respectively measure perceptual similarity, structural alignment, and signal fidelity. To assess efficiency, we also report the average inference time per image.

**Results and Analysis.**   Table 1 reports quantitative results on COCO, DarkFace, and Cartoon datasets using the SD v1.4 base model. With a tight convergence threshold ($\eta = 10^{-5}$), *PreciseInv* achieves the lowest LPIPS, highest SSIM, and best PSNR across all datasets. Despite the high quality, its inference time remains competitive. With a relaxed threshold ($\eta = 10^{-2}$), *PreciseInv* still surpasses most methods in LPIPS, SSIM, and PSNR, while reducing inference time to 5.33 s. Among forward optimization methods, *EasyInv* and *ReNoise* underperform the 1000-step *DDIM Inversion*, which suggests that numerical or iterative methods suffer from a performance bottleneck. In contrast, our method breaks through this bottleneck by a large margin. In particular, on the stylized Cartoon dataset,

Table 1: Quantitative comparison of diffusion inversion methods for image reconstruction using SD v1.4 on COCO, DarkFace, and Cartoon datasets.

| | Method | LPIPS (↓) | | | SSIM (↑) | | | PSNR (↑) | | | Time (s, ↓) |
|---|---|---|---|---|---|---|---|---|---|---|---|
| | | COCO | Dark. | Cart. | COCO | Dark. | Cart. | COCO | Dark. | Cart. | Avg. |
| Invertible Samplers | EDICT | 0.430 | 0.329 | 0.032 | 0.367 | 0.630 | 0.931 | 14.04 | 20.42 | 29.87 | 45.50 |
| | BDIA | 0.431 | 0.329 | 0.033 | 0.366 | 0.630 | 0.946 | 14.04 | 20.42 | 30.89 | 172.40 |
| | BELM | 0.431 | 0.331 | 0.041 | 0.366 | 0.617 | 0.947 | 14.03 | 20.43 | 29.97 | 9.00 |
| Forward Optimizing | DDIM Inversion | 0.118 | 0.073 | 0.145 | 0.714 | 0.871 | 0.879 | 24.67 | 30.89 | 23.50 | 41.30 |
| | ReNoise | 0.120 | 0.086 | 0.323 | 0.724 | 0.869 | 0.851 | 25.00 | 30.84 | 22.21 | 38.20 |
| | EasyInv | 0.210 | 0.210 | 0.165 | 0.666 | 0.780 | 0.883 | 23.13 | 27.60 | 21.95 | 23.60 |
| | **PreciseInv** ($\eta=10^{-2}$) | 0.104 | 0.074 | 0.025 | 0.737 | 0.861 | 0.952 | 25.13 | 31.00 | 29.71 | **5.33** |
| | **PreciseInv** ($\eta=10^{-5}$) | **0.078** | **0.052** | **0.018** | **0.756** | **0.878** | **0.960** | **25.86** | **31.79** | **31.42** | 23.29 |

Table 2: Quantitative comparison of diffusion inversion methods for image reconstruction using LCM-SD v1.5 and SDXL on the COCO dataset.

| | LCM-SD v1.5 | | | | SDXL | | | |
|---|---|---|---|---|---|---|---|---|
| | LPIPS (↓) | SSIM (↑) | PSNR (↑) | Time (s, ↓) | LPIPS (↓) | SSIM (↑) | PSNR (↑) | Time (s, ↓) |
| BELM | 0.449 | 0.362 | 13.99 | 9.1 | – | – | – | – |
| DDIM Inversion | 0.626 | 0.420 | 15.74 | 35.5 | 0.492 | 0.464 | 16.24 | 154.8 |
| ReNoise | 0.603 | 0.414 | 15.59 | 83.1 | 0.424 | 0.533 | 18.26 | 27.4 |
| EasyInv | – | – | – | – | 0.194 | 0.683 | 20.89 | 35.1 |
| **PreciseInv** ($\eta = 10^{-2}$) | 0.103 | 0.753 | 25.55 | **8.83** | 0.185 | 0.763 | 27.00 | 7.43 |
| **PreciseInv** ($\eta = 10^{-5}$) | **0.083** | **0.775** | **26.45** | 25.54 | **0.080** | **0.854** | **30.93** | 41.11 |

Table 3: Quantitative results of image reconstruction under the few-step inference setting for different inversion methods using SD v1.4 on the COCO dataset.

| | $T = 2$ | | | $T = 4$ | | |
|---|---|---|---|---|---|---|
| | LPIPS (↓) | SSIM (↑) | PSNR (↑) | LPIPS (↓) | SSIM (↑) | PSNR (↑) |
| BELM | 0.832 | 0.112 | 8.19 | 0.431 | 0.366 | 14.03 |
| DDIM Inversion | 0.331 | 0.591 | 20.62 | 0.306 | 0.593 | 20.65 |
| ReNoise | 0.199 | 0.671 | 22.92 | 0.158 | 0.702 | 23.53 |
| EasyInv | 0.563 | 0.357 | 14.97 | 0.537 | 0.332 | 13.66 |
| **PreciseInv** | **0.077** | **0.766** | **25.93** | **0.084** | **0.762** | **25.65** |

the performance gains are more pronounced, yielding lower LPIPS ($-0.127$), higher SSIM ($+0.081$), and better PSNR ($+7.92$ dB). Moreover, different invertible samplers exhibit similar reconstruction quality. Compared to *EDICT* and *BDIA*, *BELM* significantly decreases inference time from 45.5 s to 9.00 s. Although they achieve remarkable success on the Cartoon dataset, their performance remains limited on COCO and DarkFace. By comparison, our method shows robustness across data types and further improves overall efficiency.

For generality, we also present the reconstruction results on the COCO dataset using the few-step model LCM-SD v1.5 and the high-resolution model SDXL in Table 2. *PreciseInv* consistently achieves the best reconstruction quality across the two base models. With $\eta=10^{-2}$, it already significantly outperforms all baselines while maintaining fast inference (8.83 s for LCM-SD v1.5 and 7.43 s for SDXL). When using a tighter threshold ($\eta=10^{-5}$), the performance of *PreciseInv* is further improved. These results demonstrate that our method is general across different model architectures.

As shown in Table 3, we report the reconstruction performance of different inversion methods under the challenging few-step setting ($T = 2, 4$). *ReNoise* exhibits stronger few-step inference capability compared to other baselines. Nevertheless, our method achieves substantially better reconstruction quality across all metrics.

## 4.2 Prompt-driven Image Editing

Here, we present the results of *PreciseInv* on prompt-driven image editing, as shown in Fig. 3, and compare its editing performance with existing methods in Fig. 4. We demonstrate that *PreciseInv* enables the preservation of fine-grained details that are irrelevant to the editing prompt across a wide range of editing scenarios. For example, when editing a down jacket into a leather jacket, the original fabric folds are well preserved (left example of the first row in Fig. 3). Furthermore, compared to

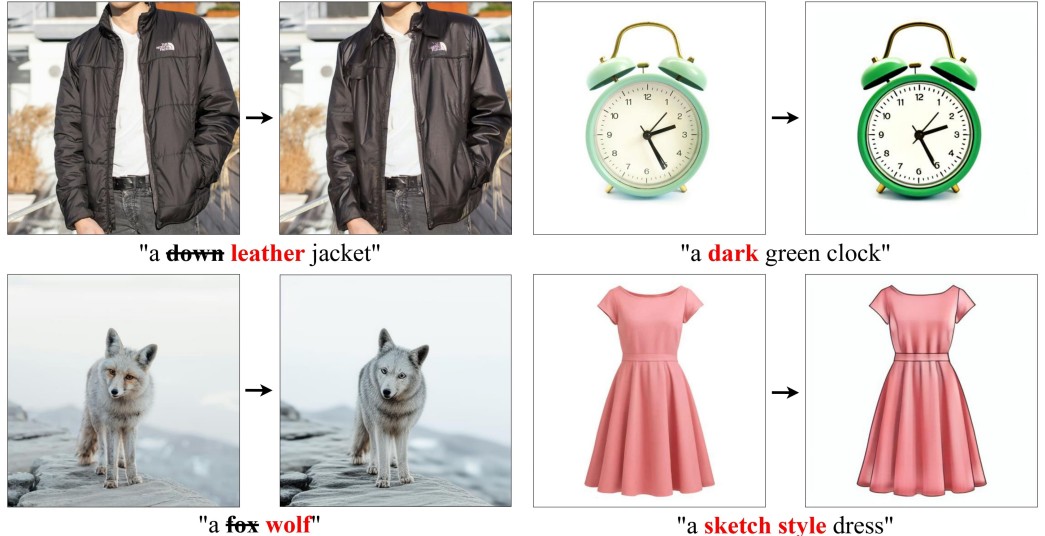

"a ~~down~~ **leather** jacket"        "a **dark** green clock"

"a ~~fox~~ **wolf**"        "a **sketch style** dress"

Figure 3: **PreciseInv** enables consistent and controllable image editing across diverse scenarios. From top left to bottom right: **(1) texture editing** (*down → leather*), **(2) color editing** (*light → dark*), **(3) object transition** (*fox → wolf*), and **(4) style transfer** (*photo → sketch*). In each case, our method preserves non-edited attributes such as identity, pose, or background, demonstrating high fidelity in semantic consistency.

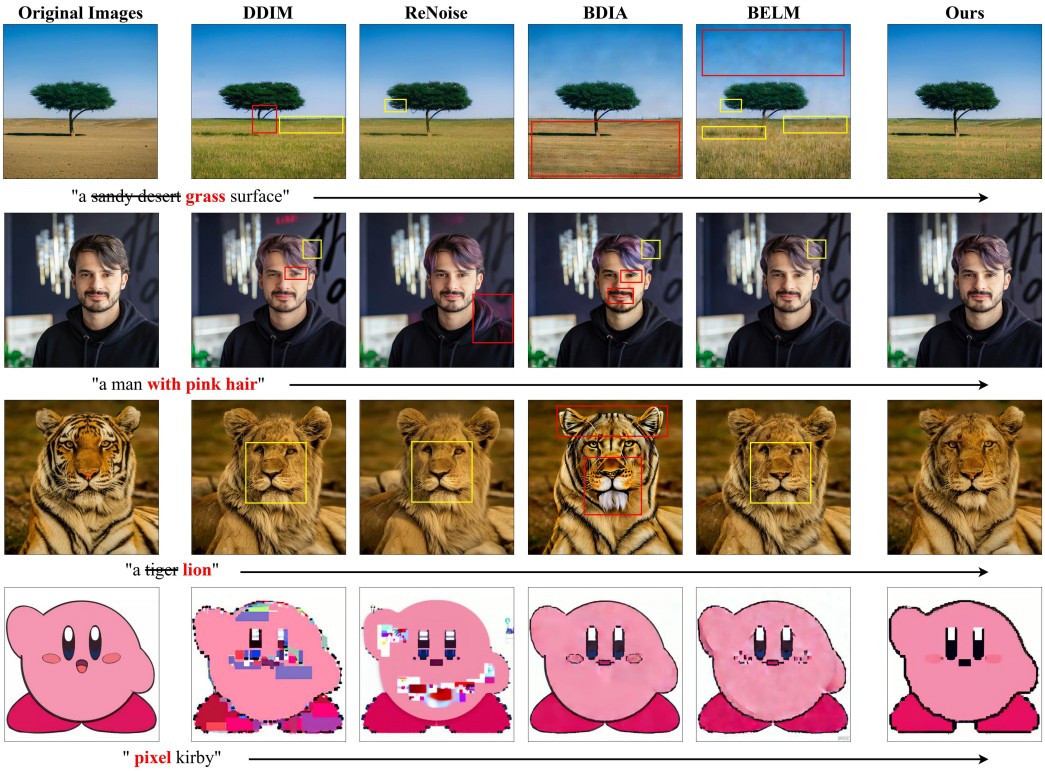

Figure 4: Comparison of different image editing results on prompt-driven image editing task.

existing methods, *PreciseInv* enables more faithful and controllable prompt-based editing. As shown in the first row of Fig. 4, our method better preserves both the tree structure and sky consistency during the texture transformation from desert to grassland, while baseline methods introduce sky artifacts (e.g., BELM) or distort tree details (e.g., ReNoise).

Notably, the purpose of these experiments is not to position *PreciseInv* as a state-of-the-art image editing method. As a generic inversion method without editing-specific design, it is inherently not comparable to advanced pipelines that involve domain-specific training [20, 40], attention manipulation [11, 16, 31, 42], or test-time optimization [7, 19]. Rather, *PreciseInv* serves as a foundation that can be integrated with such techniques. Incorporating precise inversion into editing-oriented frameworks is a promising direction for future work.

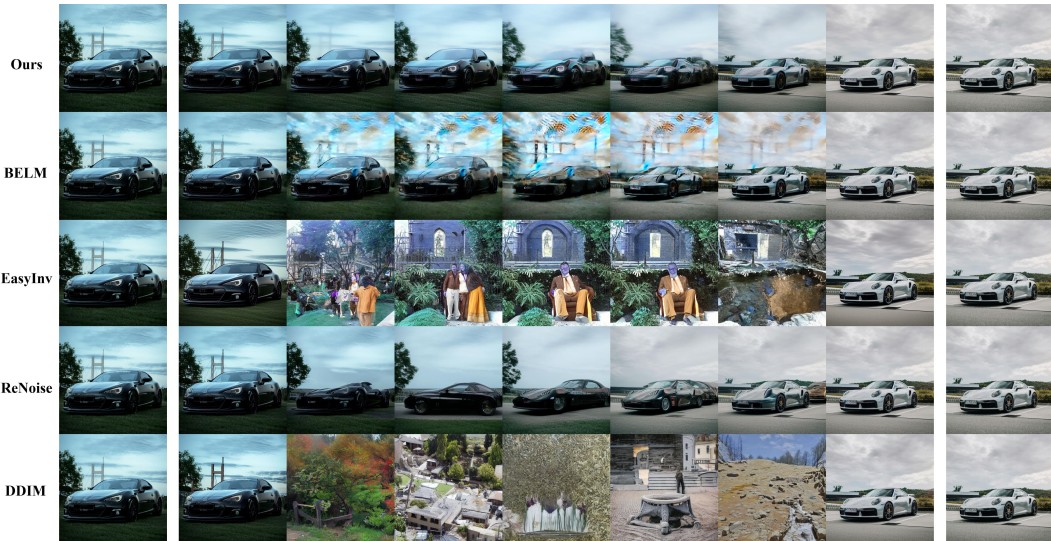

Figure 5: Interpolating between leftmost and rightmost images with spherical linear interpolation.

To further explore why *PreciseInv* enables controllable and faithful editing, we perform an interpolation-based diagnostic experiment, as shown in Fig. 5. Specifically, we first select two similar real images and apply different inversion methods to recover their corresponding latents. Then, we perform spherical linear interpolation (slerp) between the two inverted latents and reconstruct all intermediate states using the same diffusion model. As shown in Fig. 5, *PreciseInv* produces a smooth and semantically meaningful transition, maintaining structural coherence and identity consistency throughout the interpolation path. In contrast, DDIM and EasyInv often yield abrupt domain shifts or introduce foreign objects, indicating a drift away from the correct semantic manifold. BELM fails to preserve visual coherence due to accumulated noise artifacts, while ReNoise partially retains structure but suffers from geometric degradation. These findings suggest that accurate inversion leads to latents that better respect the geometry of the diffusion space, thereby enabling more localized, stable, and prompt-aligned edits.

## 5 Conclusion

We propose *PreciseInv*, a general-purpose test-time optimization framework that enables fast and faithful inversion for pre-trained diffusion models. Unlike prior fixed-point methods, we reformulate diffusion inversion as a progressive learning problem and introduce a dynamic programming-inspired strategy to solve it effectively. Theoretical analysis shows that *PreciseInv* converges under mild assumptions. Extensive experiments across diverse data domains and model architectures demonstrate that the proposed method achieves state-of-the-art performance in image reconstruction. Notably, the performance gains are especially pronounced under few-step settings and on novel samples. Moreover, we verify the effectiveness of *PreciseInv* in prompt-driven image editing tasks and further analyze its underlying mechanism. In future, we attempt to integrate *PreciseInv* with diverse editing techniques to explore a unified, controllable, and consistency-aware framework for image editing.

## Acknowledgment

This work was supported by the Hainan Provincial Joint Project of Li'an International Education Innovation Pilot Zone (Grant No.624LALH008), BUPT Kunpeng&Ascend Center of Cultivation, NSFC (Grant No.61601042), the Program for Youth Innovative Research Team of BUPT (Grant No. 2023QNTD02), and the Super Computing Platform of BUPT.

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

# Appendix & Supplementary Materials

## Contents

# A Formulations

In this section, we extend the formulation of *PreciseInv* to different combinations of base models and samplers, highlighting the generality and adaptability of our approach.

## A.1 Formulation of Stochastic Sampler: DDPM

We begin with the stochastic DDPM sampler [17]. For clarity, we first restate its sampling step:

$$\mu(\mathbf{x}_t, \mathbf{z}_t) = \frac{1}{\sqrt{\alpha_t}} \left( \mathbf{x}_t - \frac{\sqrt{1 - \alpha_t}}{\sqrt{1 - \alpha_{t-1}}} \boldsymbol{\epsilon}_\theta(\mathbf{x}_t, t) \right) + \sigma_t \mathbf{z}_t. \tag{24}$$

The added noise term $\sigma_t \mathbf{z}_t$ in Eq. (24) complicates the inversion task, as deterministic reconstruction from $\mathbf{x}_t$ is no longer guaranteed. To overcome this, we treat $\mathbf{z}_t$ as an additional learnable variable. Specifically, we modify each subproblem $\mathcal{P}(t)$ (defined in Eq. (14)) to jointly optimize the noise embedding $\boldsymbol{\epsilon}_t^*$ and the stochastic component $\mathbf{z}_t^*$:

$$\mathcal{P}(t) = \arg \min_{\boldsymbol{\epsilon}_t^*, \mathbf{z}_t^*} \|\mathcal{X}(\mathbf{x}_t^*, \mathbf{z}_t^*) - \mathbf{x}_0\|^2, \tag{25}$$

where

$$\mathbf{x}_t^* = \sqrt{\bar{\alpha}_t}\, \mathbf{x}_0 + \sqrt{1 - \bar{\alpha}_t}\, \boldsymbol{\epsilon}_t^*. \tag{26}$$

This formulation preserves the dynamic programming structure of our method. In particular, the optimal substructure property remains intact, with each subproblem $\mathcal{P}(t)$ dependent only on the solution to the previous subproblem $\mathcal{P}(t-1)$:

$$\mathcal{P}(t) = f(\mathcal{P}(t-1)) = \begin{cases} \arg \min_{\boldsymbol{\epsilon}_1^*, \mathbf{z}_1^*} \|\mu(\mathbf{x}_1^*, \mathbf{z}_1^*) - \mathbf{x}_0\|^2, & t = 1, \\ \arg \min_{\boldsymbol{\epsilon}_t^*, \mathbf{z}_t^*} \|\mu(\mathbf{x}_t^*, \mathbf{z}_t^*) - \mathbf{x}_{t-1}^*\|^2, & t > 1. \end{cases} \tag{27}$$

By learning both the latent noise and the stochastic perturbations at each step, our method generalizes to stochastic samplers and enables accurate inversion under the DDPM setting.

## A.2 Formulation of Rectified Flow Models: SD3

**Preliminary.** Rectified Flow [26] defines score-based diffusion modeling by learning a velocity field over a linear interpolation path between the data distribution $q(\mathbf{x}_0)$ and the Gaussian prior $\mathcal{N}(\mathbf{0}, \mathbf{I})$. The forward process is formulated as:

$$\mathbf{x}_t = (1 - t)\mathbf{x}_0 + t\boldsymbol{\epsilon}, \quad t \in [0, 1]. \tag{28}$$

A neural network learns a time-dependent velocity field $\mathbf{v}_\theta(\mathbf{x}_t, t)$ by minimizing the following training objective:

$$\mathcal{L}_{\mathrm{RF}}(\theta) := \mathbb{E}_{t \sim p(t),\, \mathbf{x}_0 \sim q(\mathbf{x}_0),\, \boldsymbol{\epsilon} \sim \mathcal{N}(\mathbf{0}, \mathbf{I})} \left[ \|\boldsymbol{\epsilon} - \mathbf{v}_\theta(\mathbf{x}_t, t)\|^2 \right]. \tag{29}$$

The backward (generative) process recovers $\mathbf{x}_0$ from a sample $\mathbf{x}_1 \sim \mathcal{N}(\mathbf{0}, \mathbf{I})$ by solving the following ordinary differential equation (ODE) in reverse time:

$$\frac{d\mathbf{x}_t}{dt} = \mathbf{v}_\theta(\mathbf{x}_t, t). \tag{30}$$

Unlike discrete-time diffusion models such as DDPM and DDIM, Rectified Flow does not define an explicit update rule. Instead, the reverse trajectory is obtained by numerically integrating Eq. (30), typically via Euler discretization:

$$\mathbf{x}_{t-\Delta t} \approx \mathbf{x}_t - \Delta t \cdot \mathbf{v}_\theta(\mathbf{x}_t, t). \tag{31}$$

SD3 [10] refines Rectified Flow to enable large-scale pretraining for text-to-image generation. First, instead of adopting a uniform timestep schedule, it samples timesteps from a logit-normal [3] distribution defined as:

$$\pi_{\ln}(t; m, s) = \frac{1}{s\sqrt{2\pi}} \frac{1}{t(1-t)} \exp\left( -\frac{(\mathrm{logit}(t) - m)^2}{2s^2} \right). \tag{32}$$

This design allows SD3 to flexibly control the bias and width of the timestep distribution. It further introduces a multimodal diffusion transformer that unifies cross-attention and self-attention mechanisms. This results in improved alignment between text and image modalities.

***PreciseInv* for the SD3 Model**  We further extend *PreciseInv* to the continuous-time rectified flow model, SD3. To construct a tractable denoising trajectory, we first sample a sequence of timesteps $\{t_1, t_2, \ldots, t_T\}$ from the logit-normal distribution $\pi_{\ln}(i; m, s)$ defined in Eq. (32). These sampled timesteps discretize the continuous-time trajectory into a sequence of latent variables $\{\mathbf{x}_{t_0}, \mathbf{x}_{t_1}, \ldots, \mathbf{x}_{t_T}\}$, where each $\mathbf{x}_{t_i}$ is recursively approximated by Euler integration:

$$\mathbf{x}_{t_{i-1}} = \mathbf{x}_{t_i} - (t_i - t_{i-1}) \cdot v_\theta(\mathbf{x}_{t_i}, t_i), \tag{33}$$

with $\mathbf{x}_{t_T} \sim \mathcal{N}(\mathbf{0}, \mathbf{I})$ and $i \in \{T, T-1, \ldots, 2\}$.

Following Section 3, we define $T$ subproblems $\{\mathcal{P}(t_i)\}_{i=1}^T$, each targeting a local reconstruction step along the denoising path $\mathbf{x}_{t_T} \mapsto \cdots \mapsto \mathbf{x}_{t_1}$. For each $t_i$, we parameterize the latent variable $\mathbf{x}_{t_i}^*$ with a learnable noise embedding $\boldsymbol{\epsilon}_{t_i}^*$ as:

$$\mathbf{x}_{t_i}^* = (1 - t_i)\, \mathbf{x}_0 + t_i\, \boldsymbol{\epsilon}_{t_i}^*. \tag{34}$$

Each subproblem $\mathcal{P}(t_i)$ seeks to minimize the reconstruction error between the estimated and true signal:

$$\mathcal{P}(t_i) = \arg\min_{\boldsymbol{\epsilon}_{t_i}^*} \left\| \mathcal{X}(\mathbf{x}_{t_i}^*) - \mathbf{x}_0 \right\|_2, \tag{35}$$

where $\mathcal{X}$ denotes the Euler-discretized backward trajectory from $\mathbf{x}_{t_T}$ to $\mathbf{x}_{t_1}$. For clarity, we rewrite the Euler step in Eq. (31) as:

$$\mu(\mathbf{x}_t, t_i, t_{i-1}) = \mathbf{x}_t - (t_i - t_{i-1}) \cdot \mathbf{v}_\theta(\mathbf{x}_t, t_i). \tag{36}$$

With this notation, each subproblem can be reformulated recursively as:

$$\mathcal{P}(t_i) = \begin{cases} \arg\min_{\boldsymbol{\epsilon}_{t_1}^*} \left\| \mu(\mathbf{x}_{t_1}^*, t_1, 0) - \mathbf{x}_0 \right\|^2, & i = 1, \\ \arg\min_{\boldsymbol{\epsilon}_{t_i}^*} \left\| \mu(\mathbf{x}_{t_i}^*, t_i, t_{i-1}) - \mathbf{x}_{t_{i-1}}^* \right\|^2, & i > 1. \end{cases} \tag{37}$$

By progressively solving these subproblems from $\mathcal{P}(t_1)$ to $\mathcal{P}(t_T)$, *PreciseInv* achieves a precise inversion of the image $\mathbf{x}_0$ into the SD3 latent space $\mathbf{x}_1$.

# B Proofs

We provide formal proofs for the theoretical results presented in Section 3.

## B.1 Proof of Theorem 1

*Proof.* For the function $\mathcal{L}_t(\mathbf{x}_t)$ with $L$-Lipschitz continuous gradients (Assumption 2), we define the gradient-based update as:

$$\mathbf{x}_t^{(k+1)} = \mathbf{x}_t^{(k)} - \gamma \nabla_{\mathbf{x}_t} \mathcal{L}_t(\mathbf{x}_t^{(k)}), \tag{38}$$

where $k$ denotes the iteration index and $\gamma$ is the learning rate. By the smoothness lemma [15], for any $\mathbf{x}$ and $\mathbf{x}'$,

$$f(\mathbf{x}') \leq f(\mathbf{x}) + \langle \nabla f(\mathbf{x}), \mathbf{x}' - \mathbf{x} \rangle + \tfrac{L}{2} \|\mathbf{x}' - \mathbf{x}\|^2, \tag{39}$$

where $f$ denotes any function with $L$-Lipschitz continuous gradients. Then, when $\gamma < 2/L$, we obtain

$$\mathcal{L}_t(\mathbf{x}_t^{(k+1)}) \leq \mathcal{L}_t(\mathbf{x}_t^{(k)}) - \gamma \left(1 - \tfrac{L\gamma}{2}\right) \|\nabla \mathcal{L}_t(\mathbf{x}_t^{(k)})\|^2. \tag{40}$$

After $K$ iterations, we have

$$\min_{k \in [1, K]} \|\nabla \mathcal{L}_t^{(k)}\|^2 \leq \mathcal{O}\left(\tfrac{1}{K}\right), \tag{41}$$

which implies that $\mathcal{L}_t$ converges to a stationary point $\delta_t$ during the optimization of $\mathcal{P}(t)$. □

## B.2 Proof of Theorem 2

*Proof.* Let the accumulated reconstruction error of $\mathcal{P}(t)$ be $\Gamma(t) := \|\mathcal{X}(\mathbf{x}_t) - \mathbf{x}_0\|^2$, where $\mathbf{x}_t$ denotes the estimated latent representation and $\mathbf{x}_0$ is the ground-truth image. We define $\mathcal{X}(\mathbf{x}_0) = 0$. From Theorem 1, for all $t \in \{1, \ldots, T\}$, we have:

$$\exists \, \delta_t > 0, \quad \mathcal{L}_t(\mathbf{x}_t) \le \delta_t. \tag{42}$$

Considering Eq. (18) and Assumption 1, we obtain:

$$\Gamma(t) \le \sum_{t=1}^{T} \left( \prod_{j=1}^{t-1} L_m^{(j)} \right) \delta_t. \tag{43}$$

Under Assumption 1, there exists a constant $L_m < 1$ such that $L_m^{(j)} \le L_m$. Hence,

$$\Gamma(t) \le L_m \, \Gamma(t-1) + \delta_t. \tag{44}$$

Let $\delta = \max\{\delta_1, \ldots, \delta_T\}$. By recursively applying Eq. (44) from $t = 1$ to $T$, we derive:

$$\begin{cases} \Gamma(1) \le \delta, \\ \Gamma(2) \le L_m \delta + \delta, \\ \quad \vdots \\ \Gamma(T) \le \delta \sum_{k=1}^{T} L_m^{k-1} = \delta \cdot \dfrac{1 - L_m^T}{1 - L_m} \le \dfrac{\delta}{1 - L_m}. \end{cases} \tag{45}$$

Thus, $\Gamma(T)$ is bounded by $\delta/(1 - L_m)$. $\qquad \square$

# C Implementation Details

## C.1 Training Details

We implement our method using PyTorch and the Hugging Face `diffusers` library. All diffusion model weights are frozen during training, and only the per-timestep noise embeddings $\epsilon_t^*$ are optimized via gradient descent. Following *EasyInv*, we initialize the value of $\epsilon_t^*$ by performing a single step of *DDIM Inversion*. We use the AdamW optimizer with $\beta_1 = 0.9$, $\beta_2 = 0.999$, and a weight decay of $0.01$. The learning rate is set to 0.1 for SD v1.4, 0.05 for both LCM-SD v1.5 and SDXL, and 0.025 for SD3. We use the DDIM sampler unless otherwise specified; for SD3, the Euler discrete sampler is adopted due to architectural compatibility. We keep the number of inversion steps equal to the number of sampling steps, denoted as $T$. All experiments are conducted under mixed precision: `float16` is used for SD v1.4, LCM-SD v1.5, and SDXL, while `bfloat16` is used for SD3 to ensure numerical stability. We run all experiments on a single RTX 4090 24GB GPU, except for SD3, which requires a single A100 40GB GPU due to its higher memory and compute demands.

## C.2 Hyperparameters for Image Reconstruction

As described in Section 4, we tune all methods for optimal reconstruction quality. We set the number of inference steps $T$ as follows: $T{=}1000$ for *DDIM Inversion*, $T{=}50$ for *ReNoise* and *BDIA*, $T{=}100$ for *EasyInv* and *BELM*, and $T{=}2$ for *EDICT* and our *PreciseInv*. For *ReNoise*, we disable the editing loss by setting $\lambda{=}0$ and sweep the renoising steps $n \in \{1, 2, \ldots, 9\}$. *EasyInv*, *EDICT*, *BDIA*, and *BELM* are used with default configurations from their respective papers. For *PreciseInv*, we set the convergence threshold $\eta$ to $10^{-2}$, $10^{-3}$ and $10^{-5}$, respectively. To eliminate the effect of text conditioning, all methods are evaluated with empty prompts and guidance scale set to zero.

## C.3 Pseudocode of *PreciseInv*

We present the pseudocode of the proposed *PreciseInv*. Specifically, three cases are covered: diffusion models using the DDIM sampler, diffusion models using the DDPM sampler, and rectified flow models using the Euler discrete sampler, as shown in Algorithm 2, 3, and 4, respectively.

**Algorithm 2** *PreciseInv* for Diffusion Models with DDIM Sampler
___
1: **Input:** Real image $\mathbf{x}_0$, diffusion model $\epsilon_\theta$, convergence threshold $\eta$, number of inference steps $T$
2: **Output:** Inverted latent $\mathbf{x}_T^*$
3: **for** $t = 1$ to $T$ **do**
4:      Initialize $\epsilon_t^* \sim \mathcal{N}(\mathbf{0}, \mathbf{I})$
5:      **if** $t > 1$ **then**
6:          $\mathbf{x}_{t-1}^* \leftarrow \sqrt{\bar{\alpha}_{t-1}}\, \mathbf{x}_0 + \sqrt{1 - \bar{\alpha}_{t-1}}\, \epsilon_{t-1}^*$
7:      **else**
8:          $\mathbf{x}_{t-1}^* \leftarrow \mathbf{x}_0$
9:      **end if**
10:     // Apply a single DDIM Inversion step
11:     $\epsilon_t^* \leftarrow \frac{1}{\sqrt{1-\bar{\alpha}_t}}\left(\mu(\mathbf{x}_{t-1}^*) - \sqrt{\bar{\alpha}_t}\mathbf{x}_0\right)$     ($\mu$ defined in Eq. (16))
12:     $\mathbf{x}_t^* \leftarrow \sqrt{\bar{\alpha}_t}\, \mathbf{x}_0 + \sqrt{1 - \bar{\alpha}_t}\, \epsilon_t^*$
13:     $\mathcal{L}_{\text{rec}} \leftarrow \|\mu(\mathbf{x}_t^*) - \mathbf{x}_{t-1}^*\|^2$
14:     **while** $\mathcal{L}_{\text{rec}} < \eta$ **do**
15:         $\mathbf{x}_t^* \leftarrow \sqrt{\bar{\alpha}_t}\, \mathbf{x}_0 + \sqrt{1 - \bar{\alpha}_t}\, \epsilon_t^*$
16:         $\mathcal{L}_{\text{rec}} \leftarrow \|\mu(\mathbf{x}_t^*) - \mathbf{x}_{t-1}^*\|^2$
17:         $\epsilon_t^* \leftarrow \epsilon_t^* - \nabla\mathcal{L}_{\text{rec}}$
18:     **end while**
19: **end for**
20: **return** $\mathbf{x}_T^*$
___

**Algorithm 3** *PreciseInv* for Diffusion Models with DDPM Sampler
___
1: **Input:** Real image $\mathbf{x}_0$, diffusion model $\epsilon_\theta$, convergence threshold $\eta$, number of inference steps $T$
2: **Output:** Inverted latent $\mathbf{x}_T^*$, additional noise terms $\{\mathbf{z}_T^*, \mathbf{z}_{T-1}^*, \cdots, \mathbf{z}_1^*\}$
3: **for** $t = 1$ to $T$ **do**
4:      Initialize $\epsilon_t^* \sim \mathcal{N}(\mathbf{0}, \mathbf{I})$
5:      Initialize $\mathbf{z}_t^* \sim \mathcal{N}(\mathbf{0}, \mathbf{I})$
6:      **if** $t > 1$ **then**
7:          $\mathbf{x}_{t-1}^* \leftarrow \sqrt{\bar{\alpha}_{t-1}}\, \mathbf{x}_0 + \sqrt{1 - \bar{\alpha}_{t-1}}\, \epsilon_{t-1}^*$
8:      **else**
9:          $\mathbf{x}_{t-1}^* \leftarrow \mathbf{x}_0$
10:     **end if**
11:     $\epsilon_t^* \leftarrow \frac{1}{\sqrt{1-\bar{\alpha}_t}}\left(\mu(\mathbf{x}_{t-1}^*) - \sqrt{\bar{\alpha}_t}\mathbf{x}_0\right)$     ($\mu$ defined in Eq. (24))
12:     $\mathbf{x}_t^* \leftarrow \sqrt{\bar{\alpha}_t}\, \mathbf{x}_0 + \sqrt{1 - \bar{\alpha}_t}\, \epsilon_t^*$
13:     $\mathcal{L}_{\text{rec}} \leftarrow \|\mu(\mathbf{x}_t^*, \mathbf{z}_t^*) - \mathbf{x}_{t-1}^*\|^2$
14:     **while** $\mathcal{L}_{\text{rec}} < \eta$ **do**
15:         $\mathbf{x}_t^* \leftarrow \sqrt{\bar{\alpha}_t}\, \mathbf{x}_0 + \sqrt{1 - \bar{\alpha}_t}\, \epsilon_t^*$
16:         $\mathcal{L}_{\text{rec}} \leftarrow \|\mu(\mathbf{x}_t^*, \mathbf{z}_t^*) - \mathbf{x}_{t-1}^*\|^2$
17:         $\epsilon_t^* \leftarrow \epsilon_t^* - \nabla\mathcal{L}_{\text{rec}}$
18:         $\mathbf{z}_t^* \leftarrow \mathbf{z}_t^* - \nabla\mathcal{L}_{\text{rec}}$
19:     **end while**
20: **end for**
21: **return** $\mathbf{x}_T^*, \{\mathbf{z}_T^*, \mathbf{z}_{T-1}^*, \cdots, \mathbf{z}_1^*\}$
___

# D   Additional Results

## D.1   Quantitative Results.

We provide additional quantitative results of our method on the image reconstruction task. Table 4 presents the performance of *PreciseInv* with different combinations of base models and samplers, as described in Section A. The experiments run on the COCO 2017 validation set [24]. We set the number of inference steps to $T = 3$ for SD3 and $T = 2$ for the other models. The convergence threshold $\eta = 10^{-3}$ is used for all models.

**Algorithm 4** *PreciseInv* for Rectified Flow with Euler Discrete Sampler
___
1: **Input:** Real image $\mathbf{x}_0$, rectified flow model $\boldsymbol{\epsilon}_\theta$, convergence threshold $\eta$, number of inference steps $T$, scalars $s, m$
2: **Output:** Inverted latent $\mathbf{x}_T^*$
    // Initialize discrete timesteps
3: **for** $i = 1$ to $T$ **do**
4:     $t_i \leftarrow \pi_{\ln}(i; s, m)$
5: **end for**
6: **for** $i = 1$ to $T$ **do**
7:     Initialize $\boldsymbol{\epsilon}_{t_i}^* \sim \mathcal{N}(\mathbf{0}, \mathbf{I})$
8:     **if** $t > 1$ **then**
9:         $\mathbf{x}_{t_{i-1}}^* \leftarrow (1 - t_{i-1})\,\mathbf{x}_0 + t_{i-1}\,\boldsymbol{\epsilon}_{t_{i-1}}^*$
10:     **else**
11:         $\mathbf{x}_{t_{i-1}}^* \leftarrow \mathbf{x}_0$
12:     **end if**
13:     $\mathbf{x}_{t_i}^* \leftarrow (1 - t_i)\,\mathbf{x}_0 + t_i\,\boldsymbol{\epsilon}_{t_i}^*$
14:     $\mathcal{L}_{\text{rec}} \leftarrow \|\mu(\mathbf{x}_{t_i}^*, t_i, t_{i-1}) - \mathbf{x}_{t_{i-1}}^*\|^2$    ($\mu$ defined in Eq. (36))
15:     **while** $\mathcal{L}_{\text{rec}} > \eta$ **do**
16:         $\mathbf{x}_{t_i}^* \leftarrow (1 - t_i)\,\mathbf{x}_0 + t_i\,\boldsymbol{\epsilon}_{t_i}^*$
17:         $\mathcal{L}_{\text{rec}} \leftarrow \|\mu(\mathbf{x}_{t_i}^*, t_i, t_{i-1}) - \mathbf{x}_{t_{i-1}}^*\|^2$
18:         $\boldsymbol{\epsilon}_{t_i}^* \leftarrow \boldsymbol{\epsilon}_{t_i}^* - \nabla \mathcal{L}_{\text{rec}}$
19:     **end while**
20: **end for**
21: $\mathbf{x}_{t_T}^* \leftarrow (1 - t_T)\,\mathbf{x}_0 + t_T\,\boldsymbol{\epsilon}_{t_T}^*$
22: **return** $\mathbf{x}_T^*$
___

Table 4 shows that all models achieve comparable reconstruction quality. SD3 attains higher SSIM and PSNR scores, while other models perform better on LPIPS. Compared to the results reported in Tables 1 and 2, *PreciseInv* maintains state-of-the-art reconstruction performance, further demonstrating the generality and robustness of our method. However, SD3 exhibits lower inference efficiency relative to other models, which can be attributed to its larger model size and the absence of an effective initialization manner.

To address this, future work will explore a progressively decreasing timestep optimization strategy for large models such as SD3. We believe that this strategy will result in a smoother optimization landscape. Additionally, exploring better initialization techniques, such as using initial values derived from existing flow-based inversion methods, would also be a promising direction.

Table 4: Additional quantitative results for image reconstruction, including: (i) Stable Diffusion 3.5 medium model with default Euler Discrete sampler, and (ii) Stable Diffusion v1.4, Latent Consistent Model Stable Diffusion v1.5, and Stable Diffusion XL with DDPM sampler.

| Model | Sampler | LPIPS ($\downarrow$) | SSIM ($\uparrow$) | PSNR ($\uparrow$) | Time (s, $\downarrow$) |
|---|---|---|---|---|---|
| SD3 | Euler | 0.104 | 0.896 | 30.12 | 403.31 |
| SD v1.4 | DDPM | 0.080 | 0.764 | 25.93 | 9.08 |
| LCM-SD v1.5 | DDPM | 0.076 | 0.784 | 26.84 | 12.07 |
| SDXL | DDPM | 0.091 | 0.842 | 30.39 | 18.59 |

We also report reconstruction performance under varying convergence thresholds $\eta$ and inference step counts $T$ in Table 5. We randomly sample 20 images from the COCO 2017 validation set and evaluate using the Stable Diffusion v1.4 model with a DDIM sampler. The number of inference steps is varied as $T \in 2, 5, 10, 20, 50, 100$ and the convergence threshold as $\eta \in 10^{-2}, 10^{-3}, 10^{-5}, 10^{-6}, 10^{-7}$. We report metrics including LPIPS, SSIM, and PSNR, as well as inference time in seconds.

From Table 5, we draw the following observations. First, for a fixed number of inference steps $T$, decreasing the convergence threshold $\eta$ consistently improves reconstruction quality across all metrics—LPIPS decreases, while SSIM and PSNR increase. However, this comes at the cost of

significantly longer inference time, especially as $\eta$ drops below $10^{-5}$. Second, we observe diminishing returns in reconstruction quality when $\eta < 10^{-5}$, particularly for small-step regimes (e.g., $T \leq 10$). This suggests that extremely strict convergence criteria may be unnecessary when the number of inference steps is limited. Third, when $\eta$ is fixed, increasing the number of inference steps $T$ initially degrades the reconstruction quality before improving it. This phenomenon results from a trade-off: while smaller per-step initialization errors benefit from higher $T$, the total accumulated error at convergence (approximately $T \cdot \eta$) also increases. Notably, due to the smooth structure of the diffusion latent space, error accumulation is sublinear and does not dominate overall performance. Fourth, inference time exhibits a non-monotonic trend with respect to $T$. When $\eta \geq 10^{-5}$, increasing $T$ initially reduces inference time—likely due to faster convergence from smaller per-step errors—but then increases it again as the step count becomes the dominant factor. In contrast, for stricter thresholds ($\eta \leq 10^{-5}$), inference time generally decreases with larger $T$, implying that the time saved through easier optimization outweighs the overhead introduced by additional steps. Finally, when $T \leq 10$, tightening the threshold from $\eta = 10^{-5}$ to $\eta = 10^{-6}$ yields negligible gains in reconstruction quality. Moreover, for large-step configurations ($T > 10$), using $\eta = 10^{-7}$ achieves reconstruction performance on par with small-step setups at $\eta = 10^{-5}$. Taken together, these findings suggest that setting $\eta = 10^{-5}$ and $T = 2$ strikes a favorable balance between efficiency and quality.

Table 5: Image reconstruction results under varying convergence thresholds $\eta$ and inference steps $T$. The table reports four metrics: LPIPS ($\downarrow$), SSIM ($\uparrow$), PSNR ($\uparrow$), and inference time in seconds ($\downarrow$).

| Metric | $\eta$ | $T = 2$ | $T = 5$ | $T = 10$ | $T = 20$ | $T = 50$ | $T = 100$ |
|---|---|---|---|---|---|---|---|
| LPIPS | $10^{-2}$ | 0.069 | 0.205 | 0.138 | 0.095 | 0.044 | 0.036 |
| | $10^{-3}$ | 0.037 | 0.052 | 0.139 | 0.096 | 0.044 | 0.036 |
| | $10^{-5}$ | 0.031 | 0.031 | 0.034 | 0.074 | 0.044 | 0.036 |
| | $10^{-6}$ | 0.031 | 0.031 | 0.031 | 0.037 | 0.045 | 0.036 |
| | $10^{-7}$ | - | - | - | 0.031 | 0.036 | 0.032 |
| SSIM | $10^{-2}$ | 0.876 | 0.839 | 0.864 | 0.877 | 0.894 | 0.896 |
| | $10^{-3}$ | 0.894 | 0.886 | 0.867 | 0.877 | 0.894 | 0.896 |
| | $10^{-5}$ | 0.897 | 0.898 | 0.896 | 0.059 | 0.895 | 0.896 |
| | $10^{-6}$ | 0.897 | 0.898 | 0.898 | 0.896 | 0.895 | 0.897 |
| | $10^{-7}$ | - | - | - | 0.898 | 0.897 | 0.898 |
| PSNR | $10^{-2}$ | 31.72 | 26.75 | 26.42 | 26.91 | 31.30 | 32.65 |
| | $10^{-3}$ | 33.12 | 29.75 | 26.15 | 26.89 | 31.30 | 32.65 |
| | $10^{-5}$ | 33.33 | 33.33 | 32.76 | 28.04 | 31.10 | 32.63 |
| | $10^{-6}$ | 33.33 | 33.35 | 33.37 | 33.03 | 30.62 | 32.63 |
| | $10^{-7}$ | - | - | - | 33.33 | 33.11 | 33.29 |
| Time (s) | $10^{-2}$ | 4.05 | 2.27 | 1.50 | 2.42 | 5.18 | 9.56 |
| | $10^{-3}$ | 8.25 | 7.13 | 2.61 | 3.85 | 5.08 | 9.82 |
| | $10^{-5}$ | 23.76 | 24.82 | 37.77 | 21.44 | 7.16 | 12.23 |
| | $10^{-6}$ | 99.60 | 82.06 | 55.77 | 68.18 | 62.99 | 26.68 |
| | $10^{-7}$ | - | - | - | 242.63 | 160.51 | 107.98 |

## D.2  Qualitative Results.

Here, we present additional qualitative results on image reconstruction, prompt-based image editing, and image interpolation.

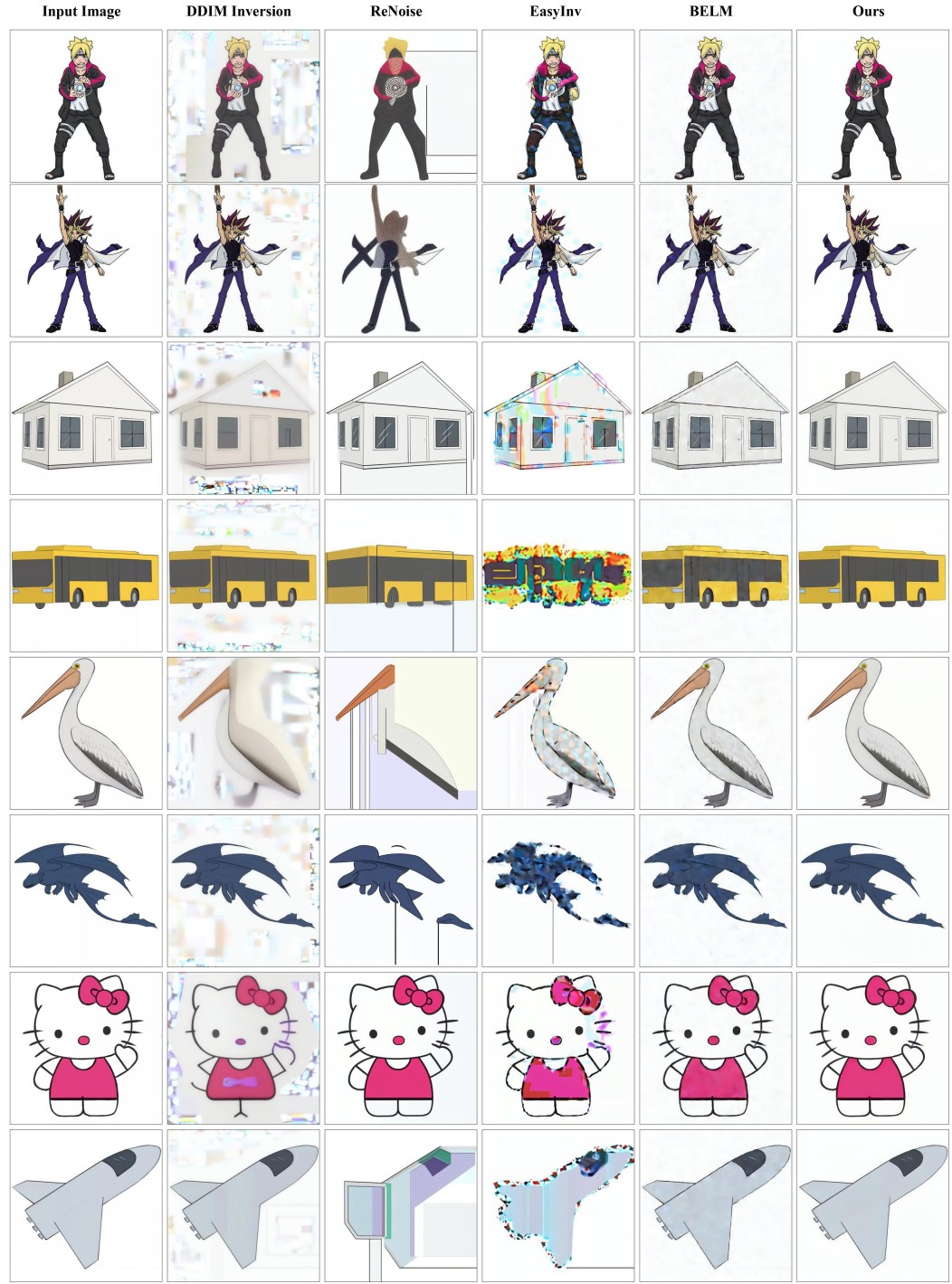

Figure 6: Qualitative comparison of image reconstruction results from different diffusion inversion methods using Stable Diffusion v1.4 on the Cartoon dataset.

Input Image   DDIM Inversion   ReNoise   EasyInv   BELM   Ours

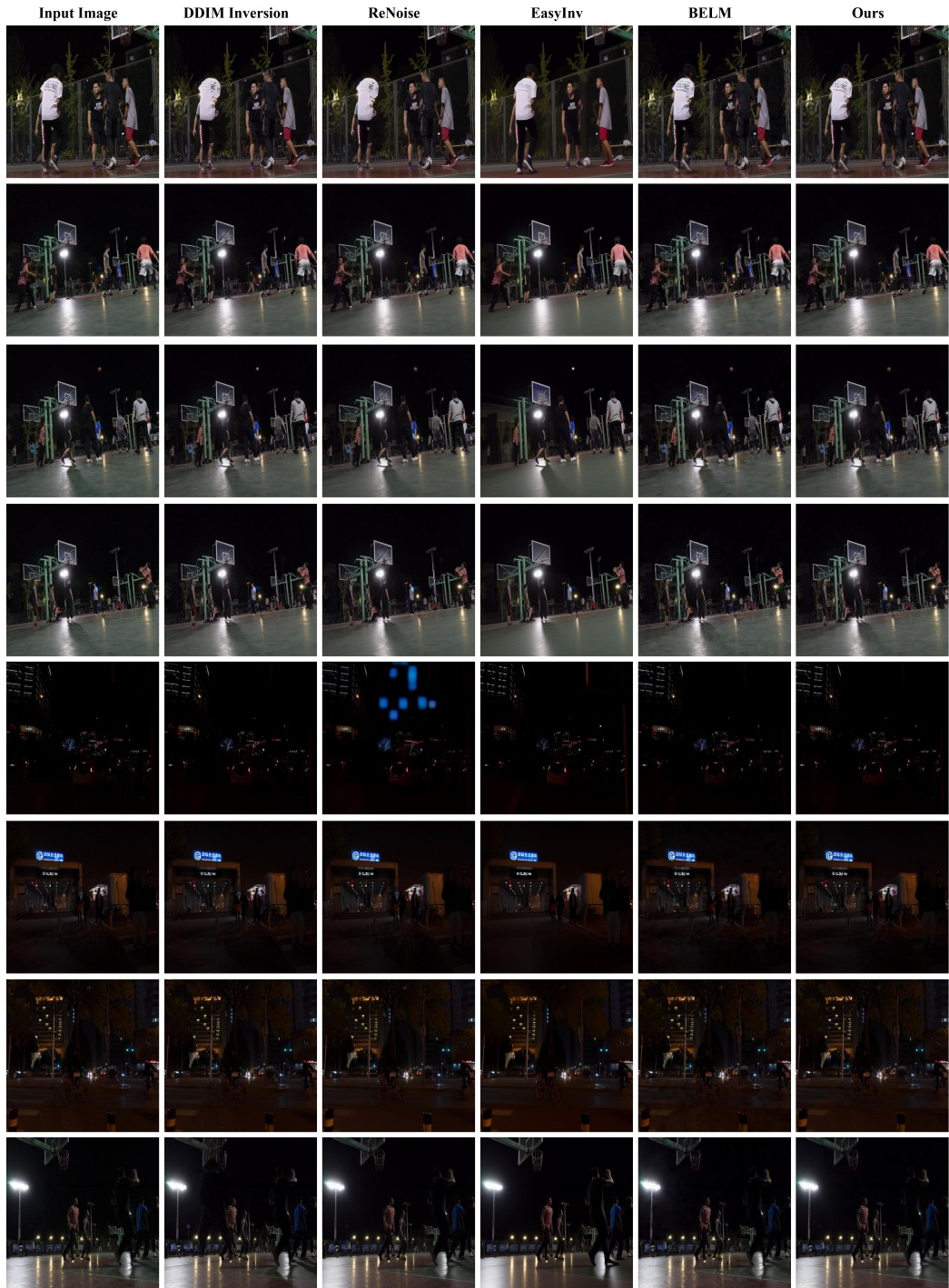

Figure 7: Qualitative comparison of image reconstruction results from different diffusion inversion methods using Stable Diffusion v1.4 on the DarkFace dataset.

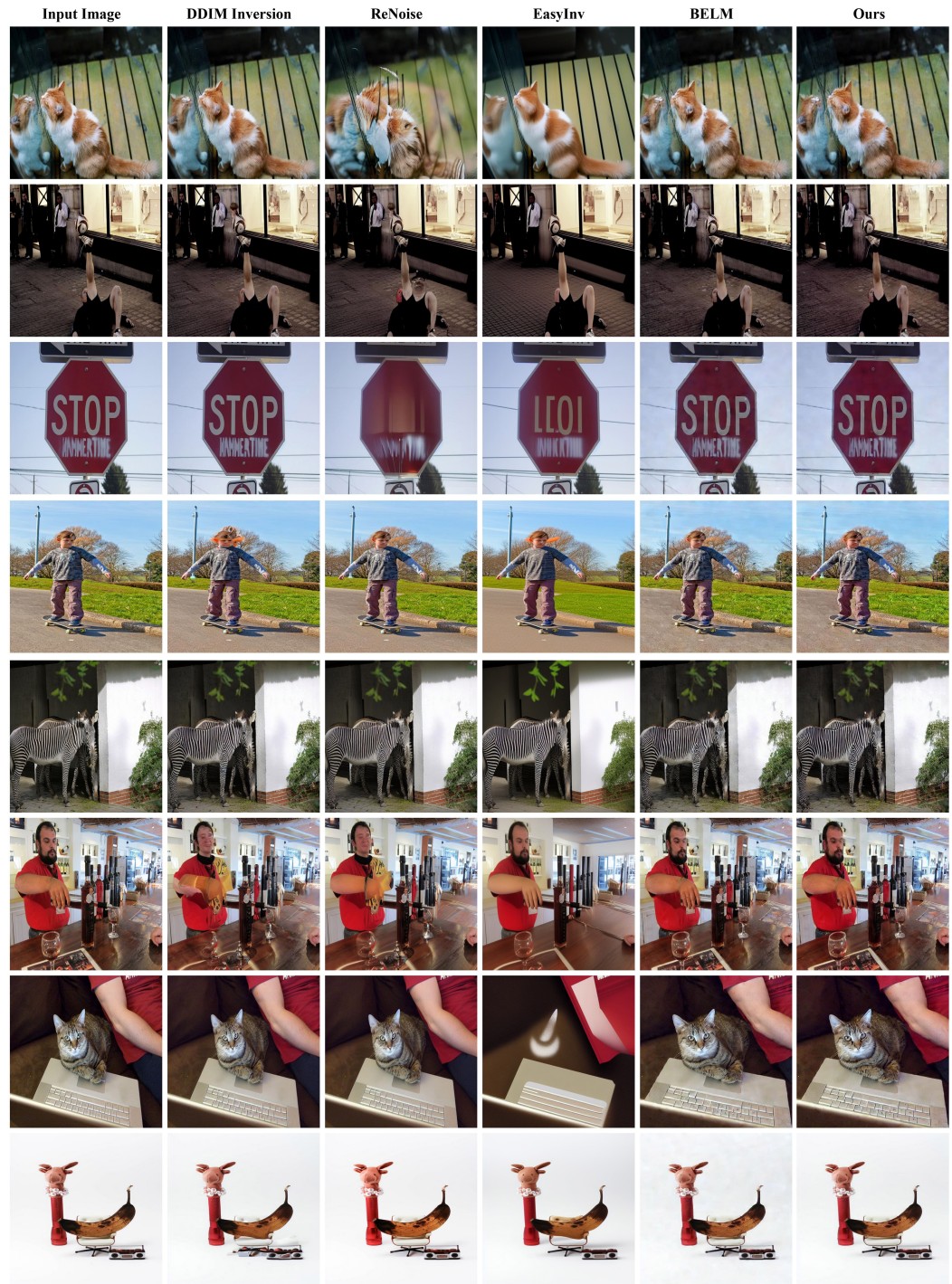

Figure 8: Qualitative comparison of image reconstruction results from different diffusion inversion methods using Stable Diffusion v1.4 on the COCO dataset.

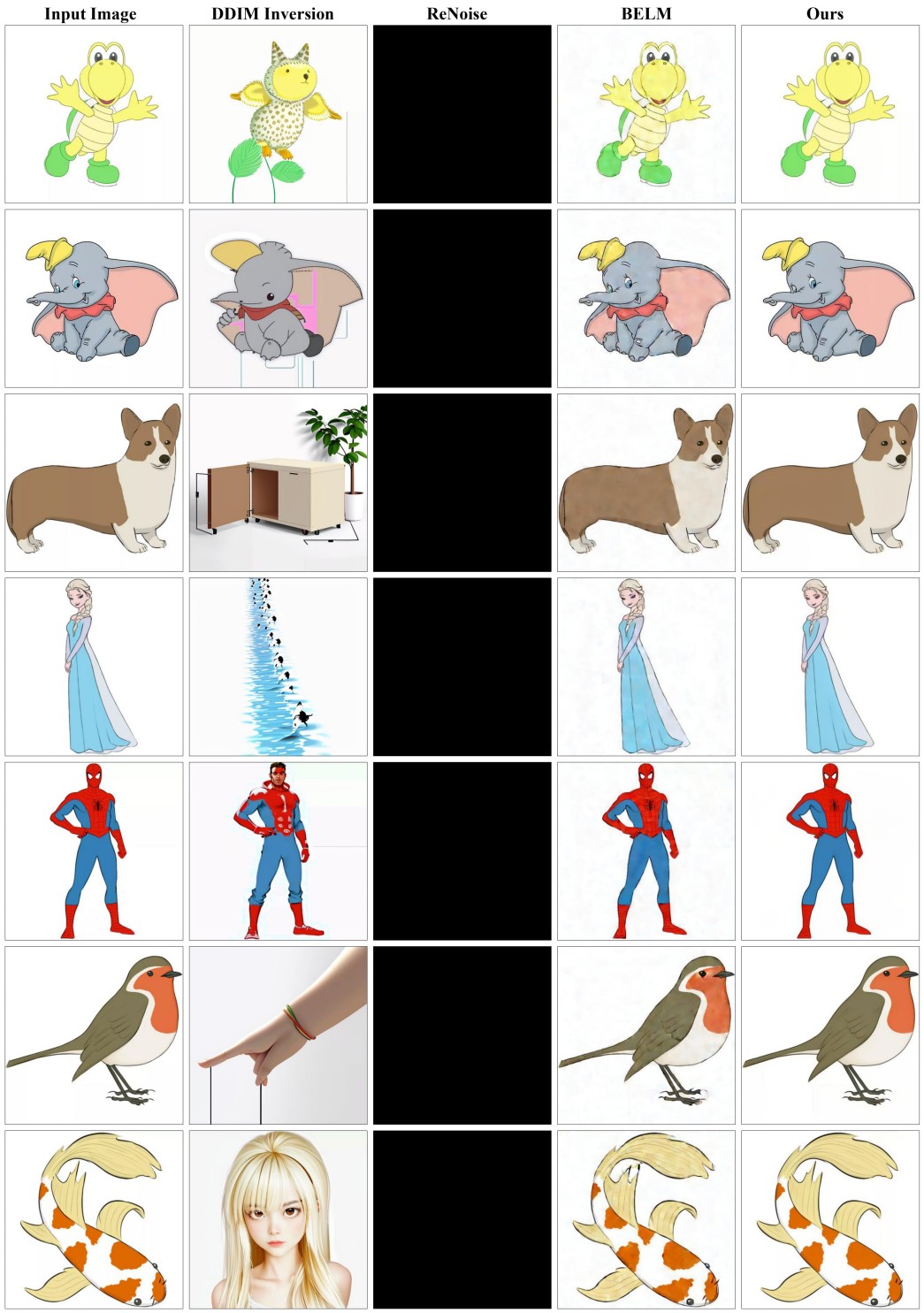

Figure 9: Qualitative comparison of image reconstruction results from different diffusion inversion methods using the Latent Consistency Model with Stable Diffusion v1.5 on the Cartoon dataset.

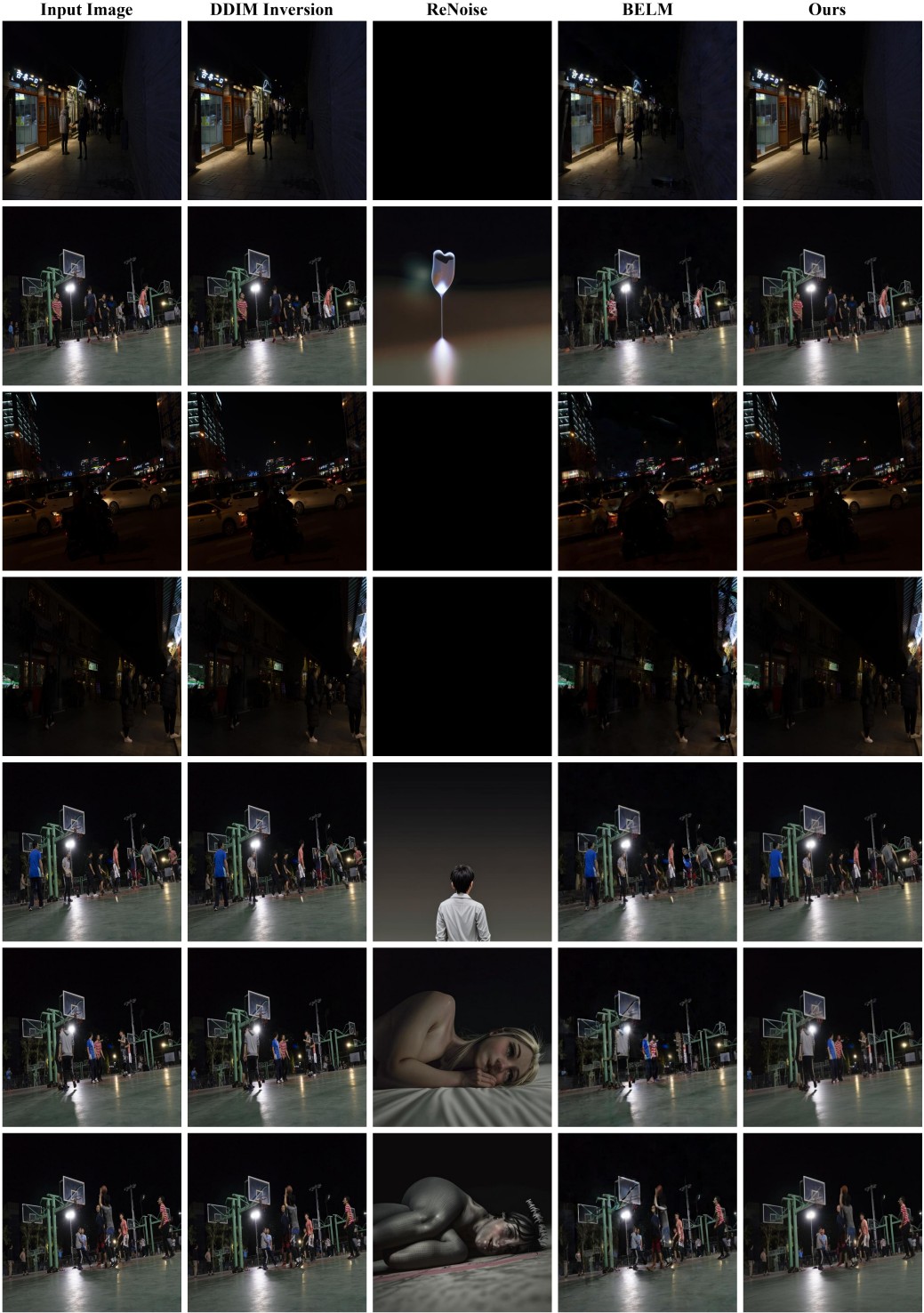

Figure 10: Qualitative comparison of image reconstruction results from different diffusion inversion methods using the Latent Consistency Model with Stable Diffusion v1.5 on the DarkFace dataset.

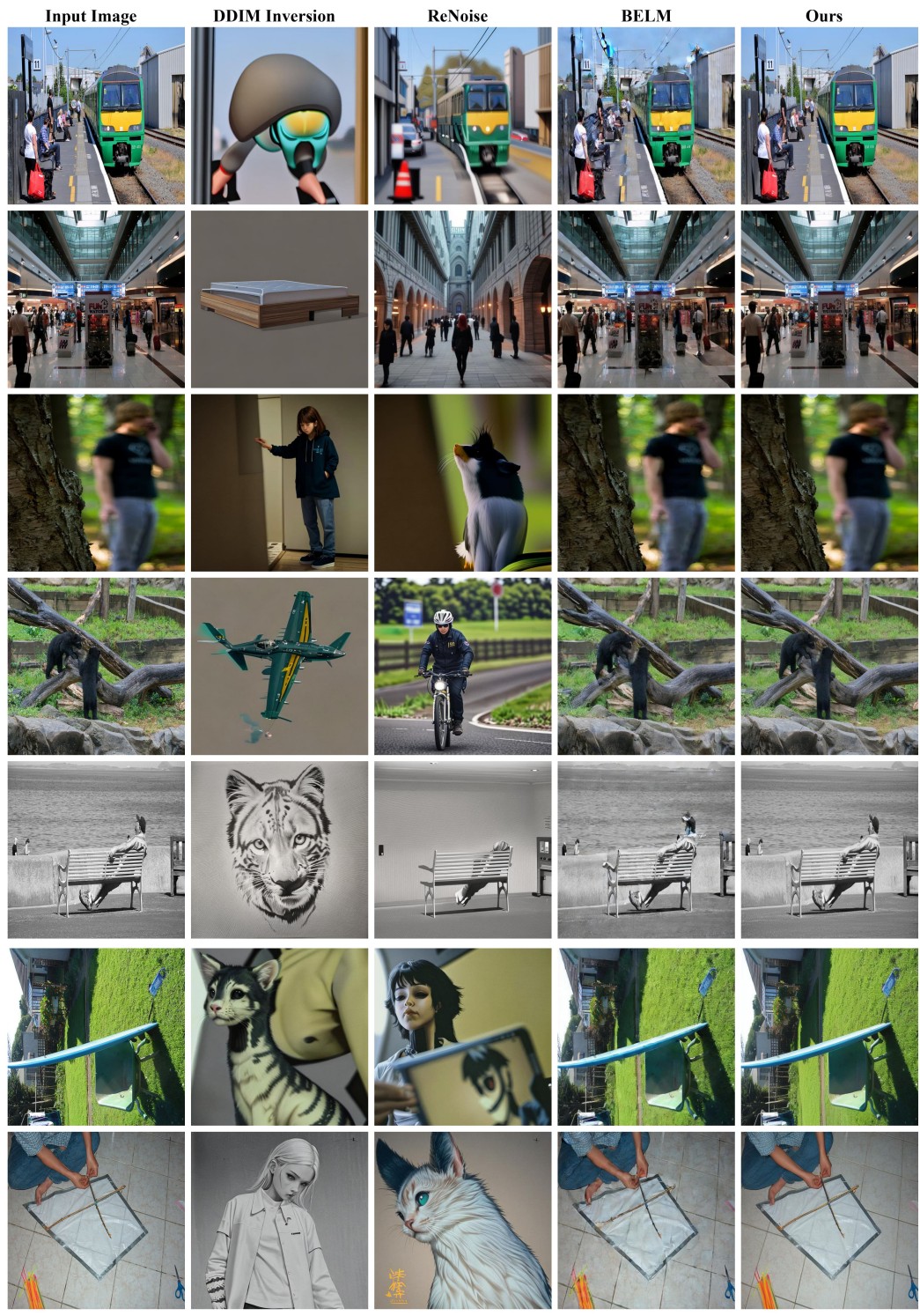

Figure 11: Qualitative comparison of image reconstruction results from different diffusion inversion methods using the Latent Consistency Model with Stable Diffusion v1.5 on the COCO dataset.

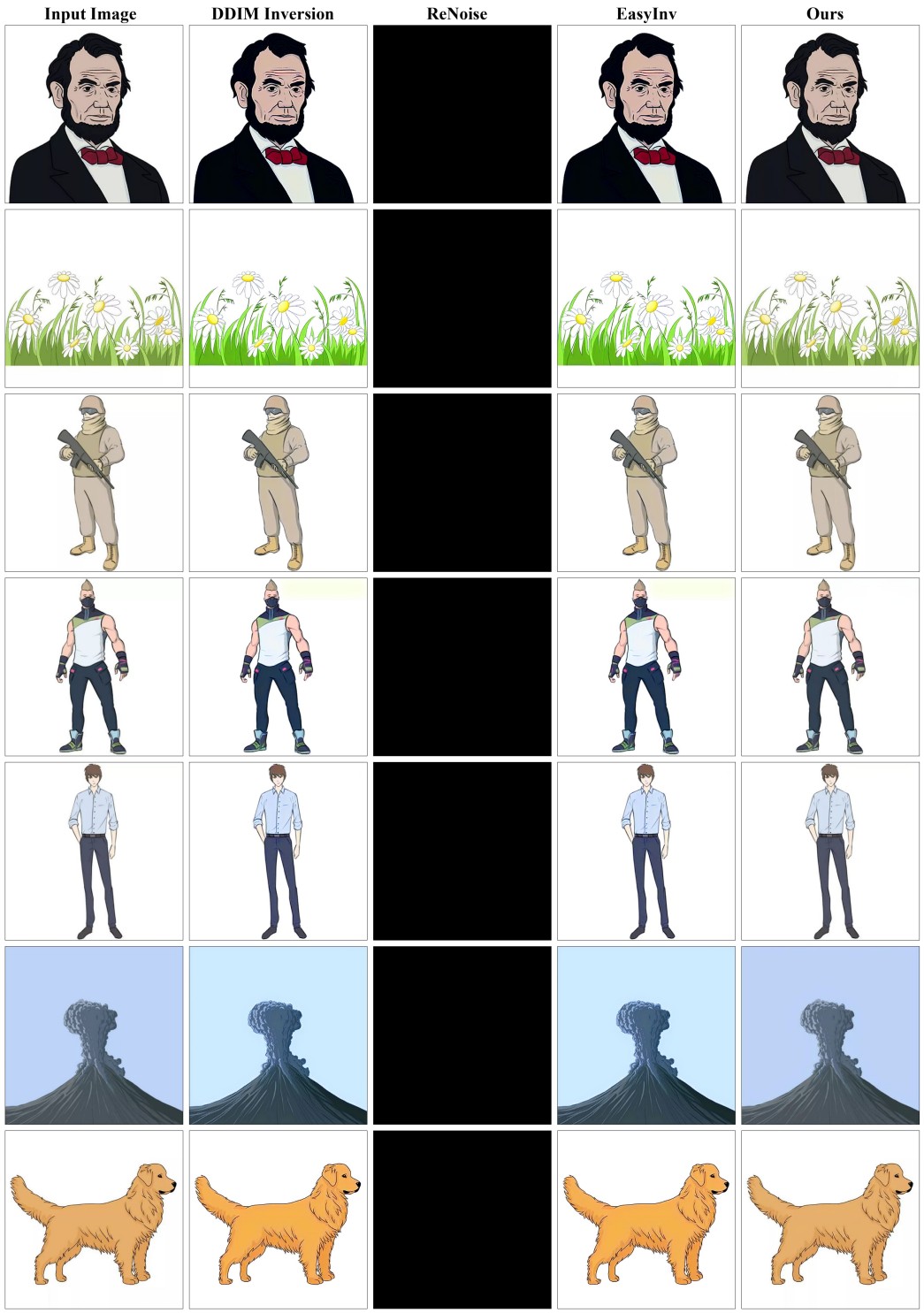

Figure 12: Qualitative comparison of image reconstruction results from different diffusion inversion methods using the Stable Diffusion XL on the Cartoon dataset.

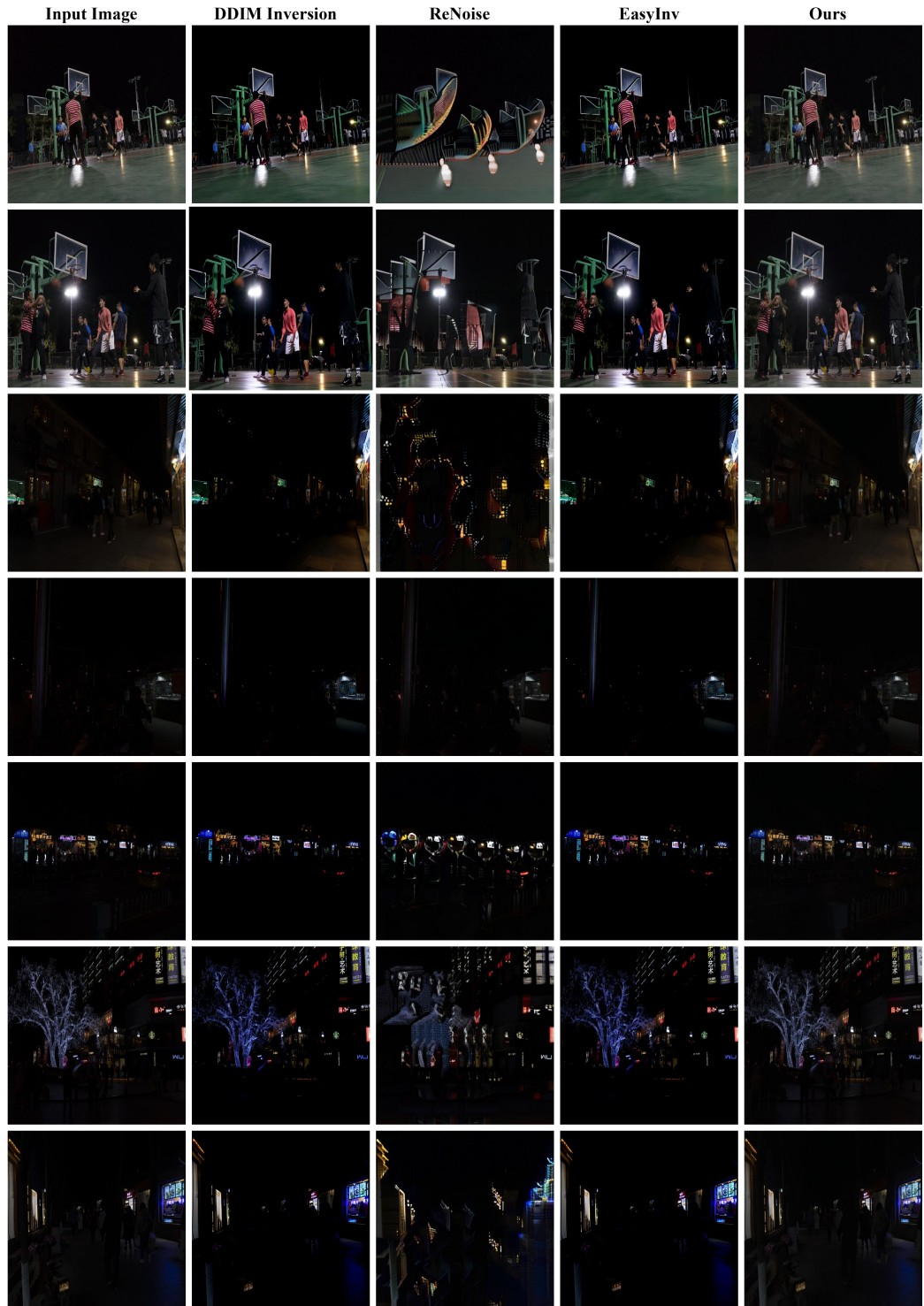

Figure 13: Qualitative comparison of image reconstruction results from different diffusion inversion methods using the Stable Diffusion XL on the DarkFace dataset.

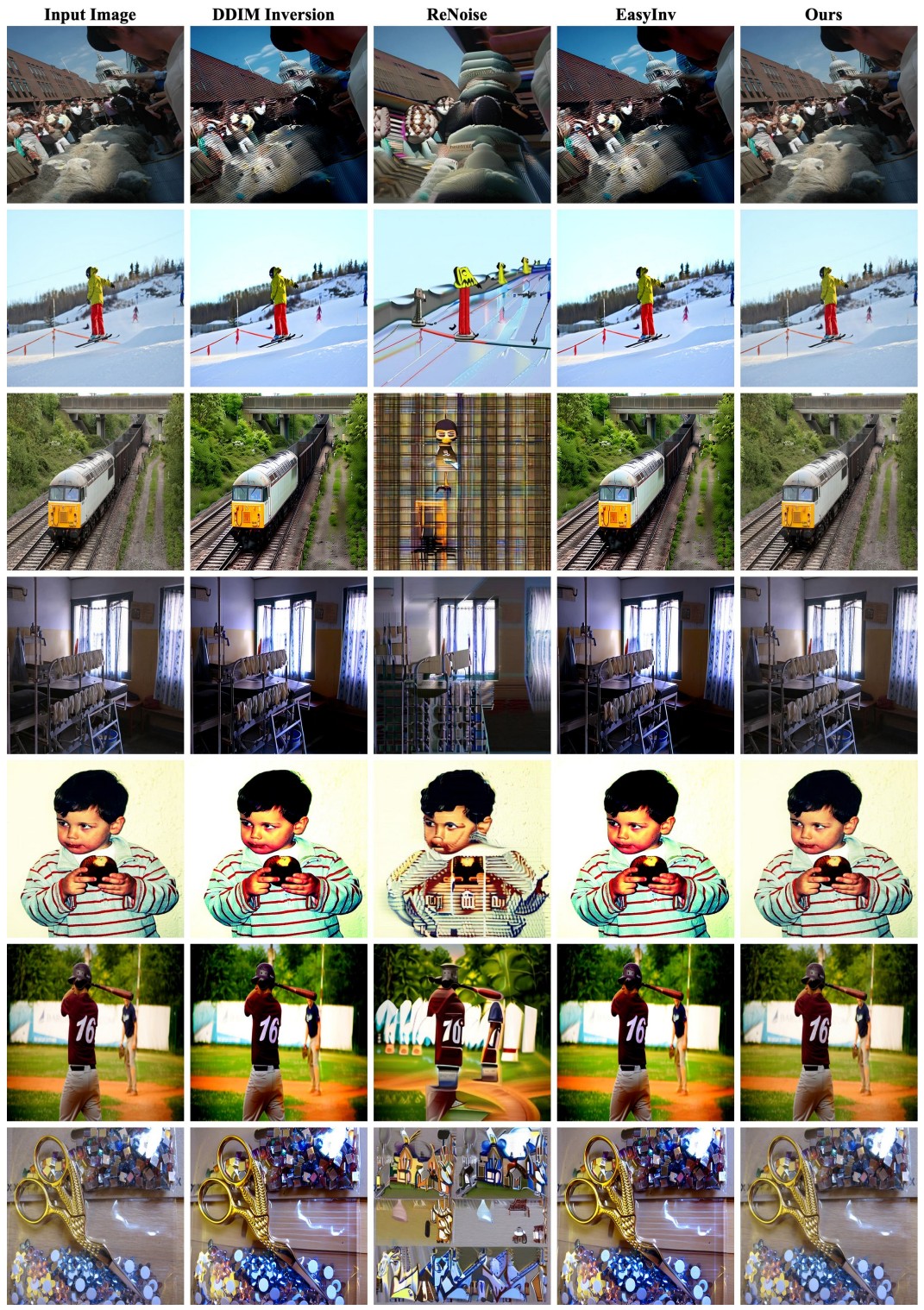

Figure 14: Qualitative comparison of image reconstruction results from different diffusion inversion methods using the Stable Diffusion XL on the COCO dataset.

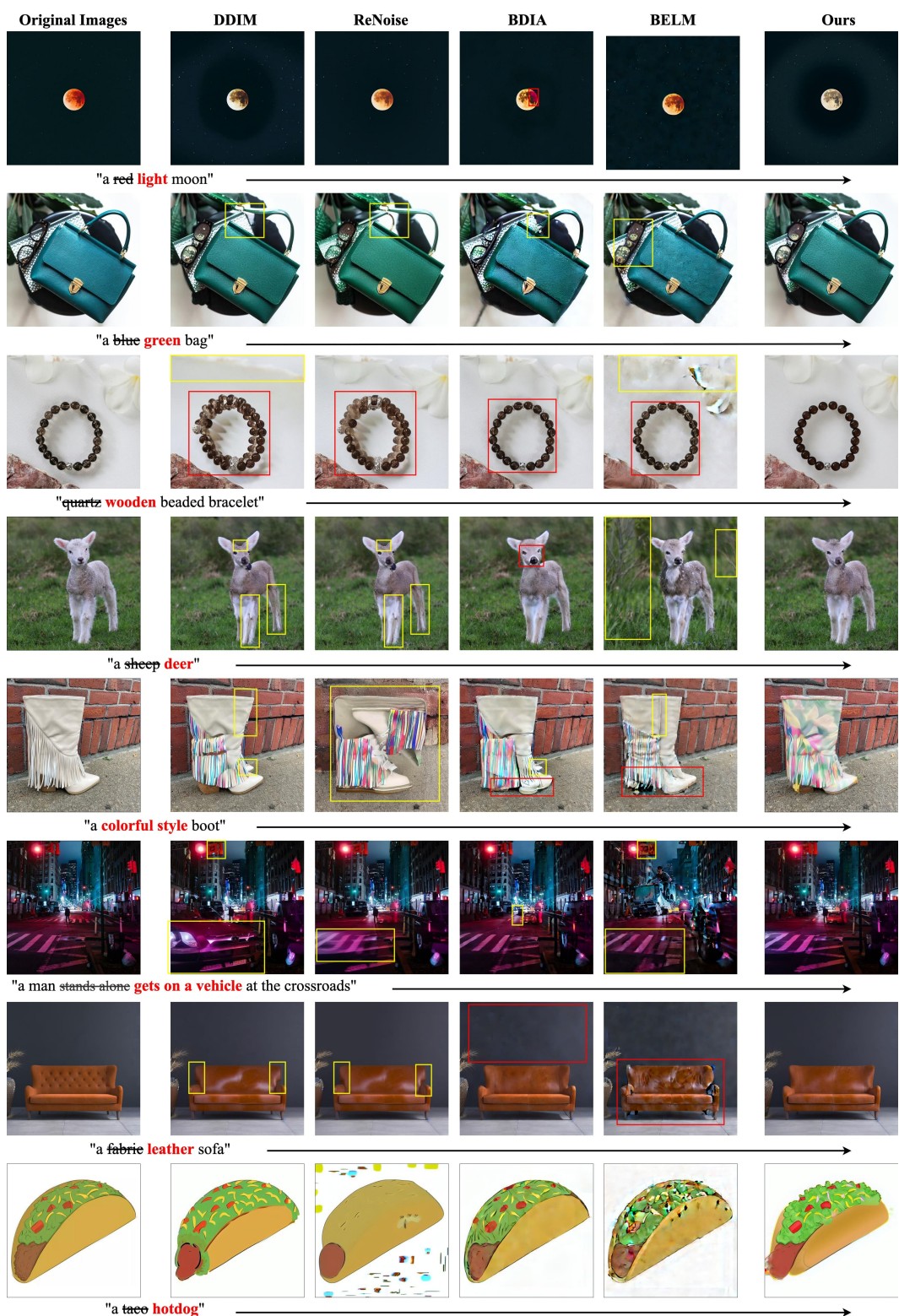

Figure 15: Qualitative comparison of prompt-driven image editing results from different diffusion inversion methods using the Stable Diffusion v1.4.

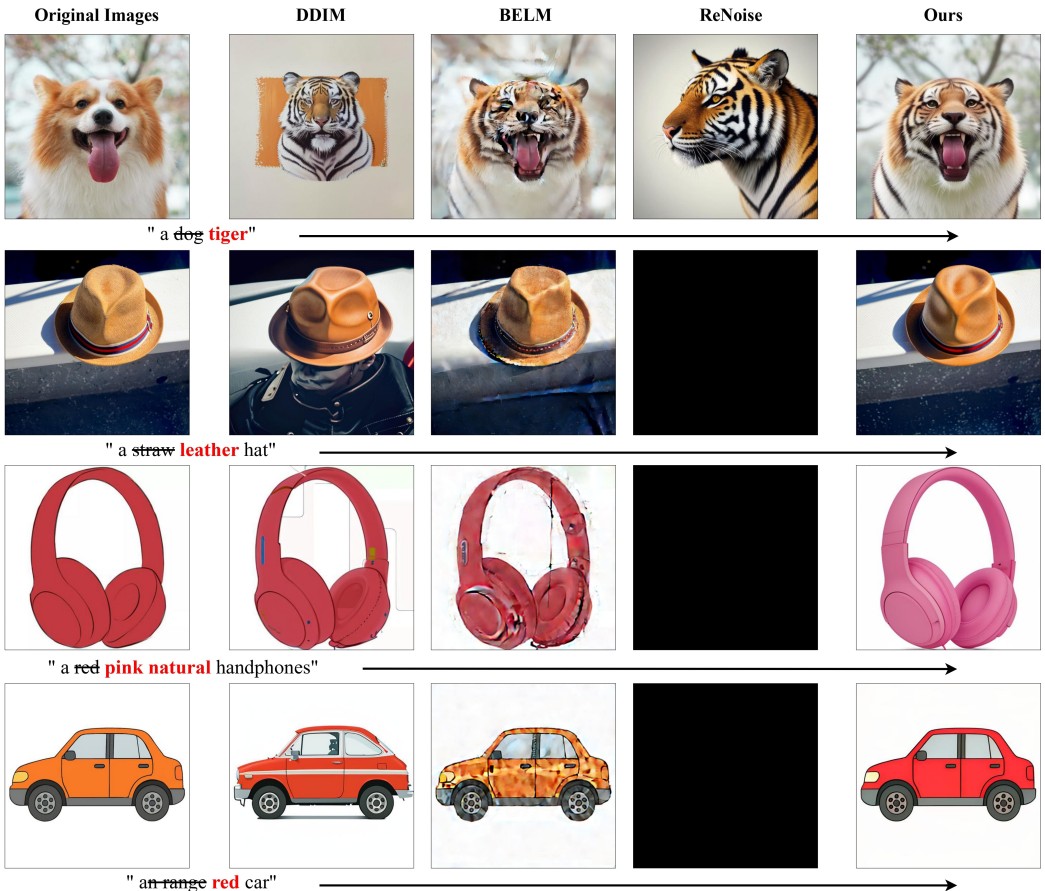

Figure 16: Qualitative comparison of prompt-driven image editing results from different diffusion inversion methods using the Latent Consistency Model with Stable Diffusion v1.5.

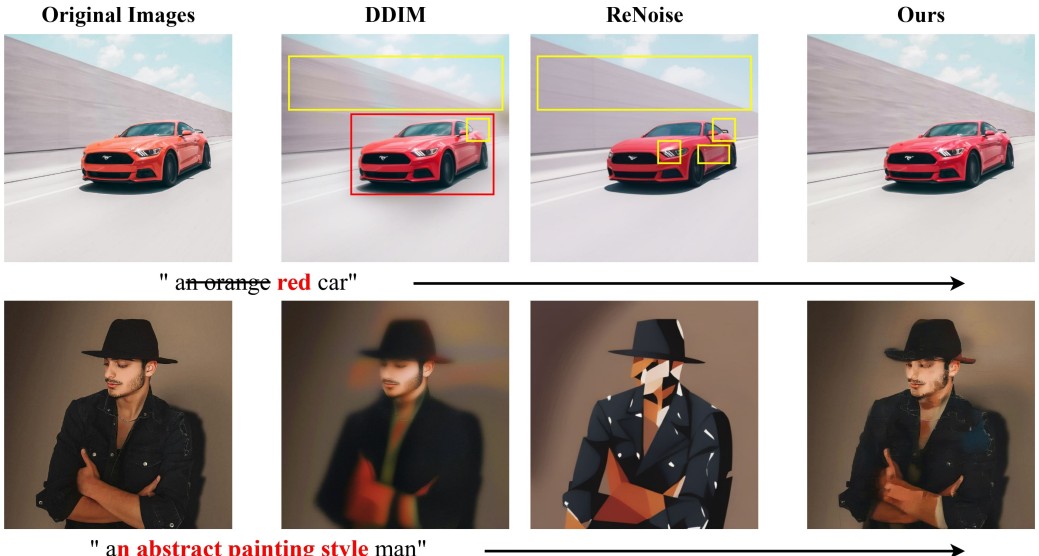

Figure 17: Qualitative comparison of prompt-driven image editing results from different diffusion inversion methods using the Stable Diffusion SDXL.

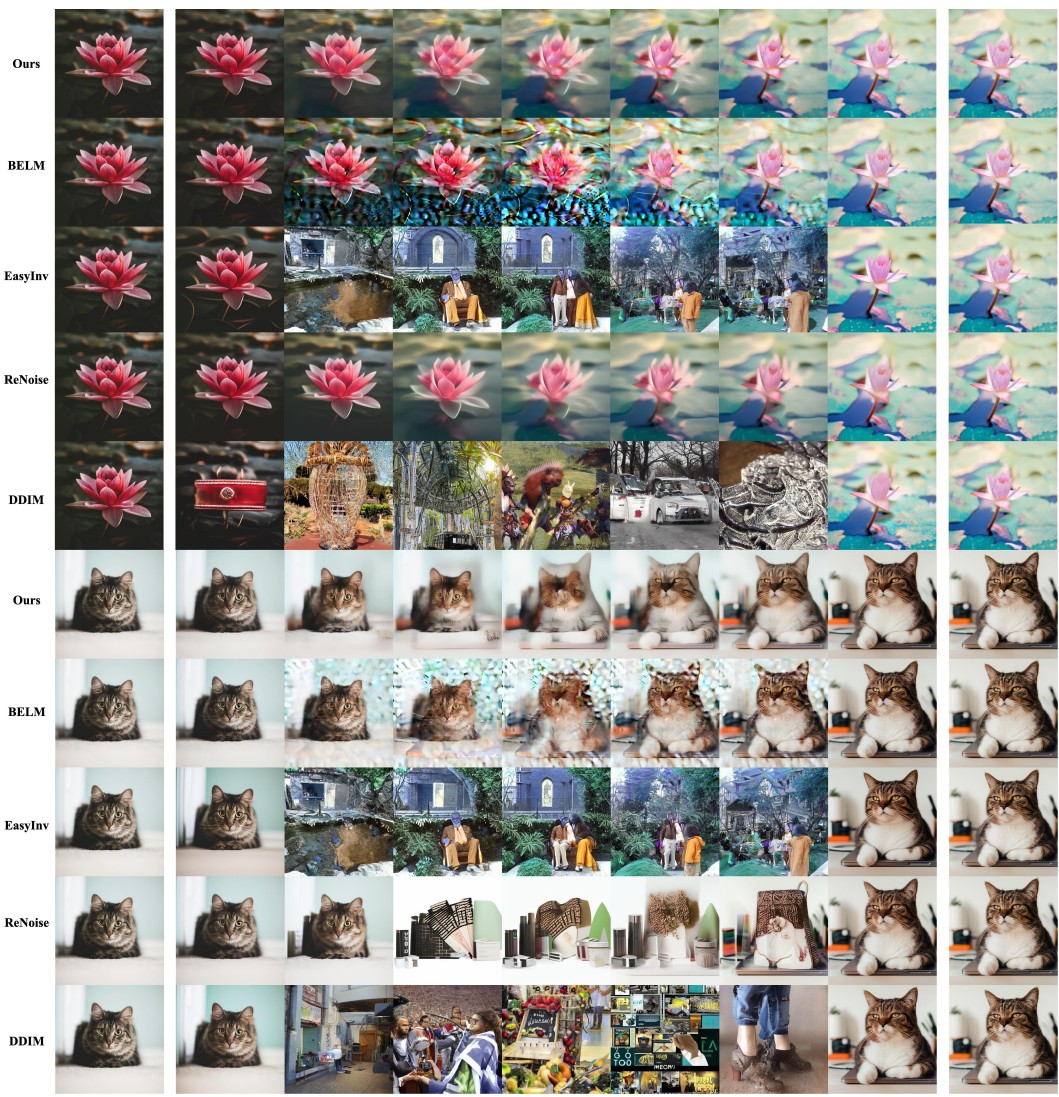

Figure 18: Qualitative comparison of image interpolation from different diffusion inversion methods using the Stable Diffusion v1.4.

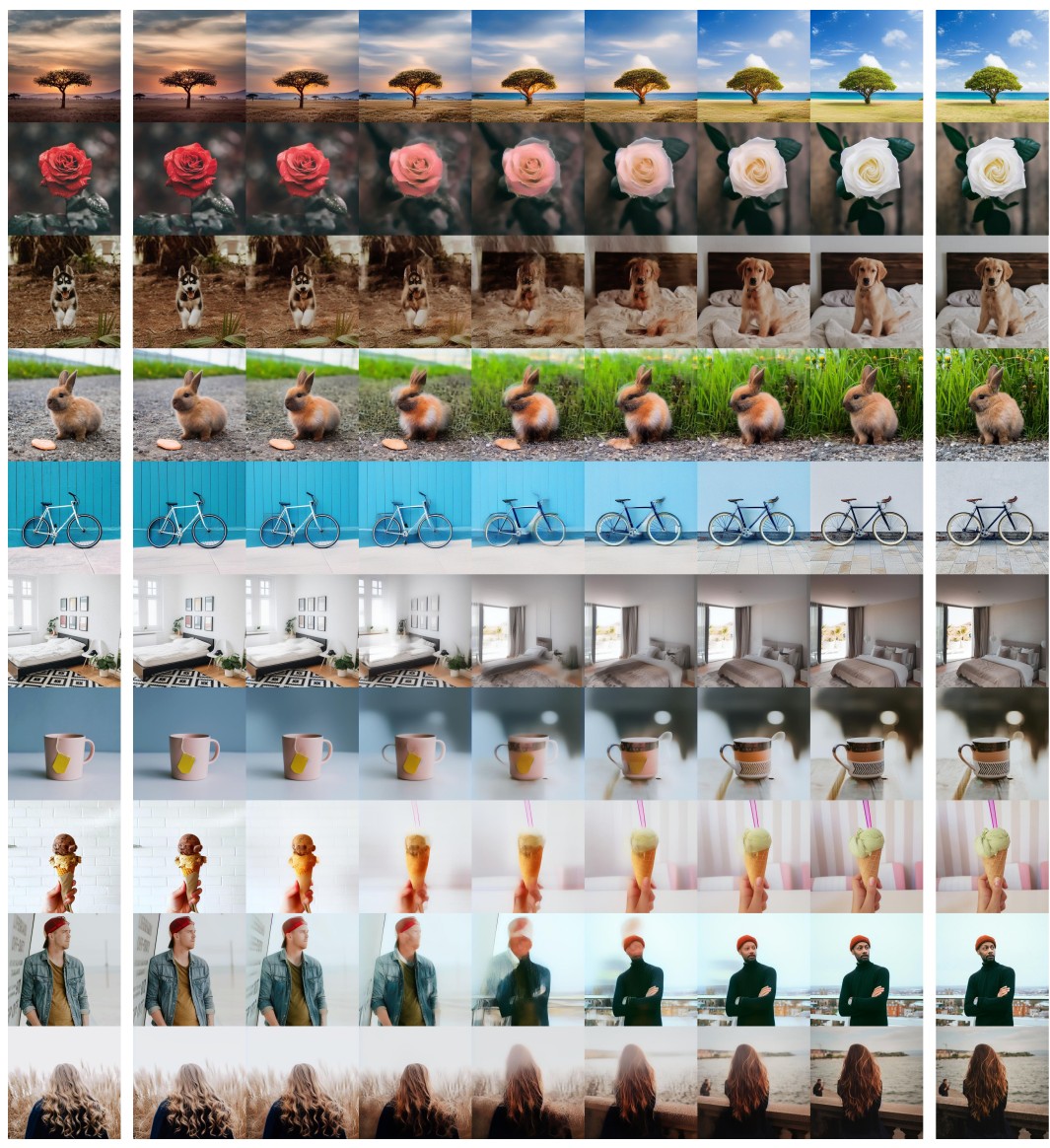

Figure 19: Additional qualitative result of image interpolation from the proposed *PreciseInv* using the Stable Diffusion v1.4.

