# OpenReview forum: "Precise Diffusion Inversion: Towards Novel Samples and Few-Step Models"
_NeurIPS.cc/2025/Conference — NeurIPS 2025 poster_

### Official Review · Reviewer_SWAE · 2025-06-26

**Clarity:** 3
**Significance:** 3
**Originality:** 3
**Rating:** 4
**Confidence:** 3

**Summary:**

This paper studies *inverse* diffusion, the goal is to find the noisy $x_T$, which has generated the provided image $x_0$. Where $x_T$ is the noise in the *latent* space of a Latent Diffusion Model. This is not a trivial task, given that the diffusion model itself is asymmetric, and simply inverting the sampler does not yield satisfactory results. In this paper inverse diffusion is seen as an optimization problem, where iteratively the optimal noise is sought, based on the optimal noise of the previous time: $x_{t-1} \rightarrow x_t$. This makes use of the Markov property of diffusion, and is solved iteratively starting from $t=1$ to $t=T$. Experimentally the method is validated on 3 different datasets and in various settings (different values for T, different LDMs), the results show that compared to related works the proposed method is advantageous (especially in LPIPS). Moreover for prompt rewriting some quantitative results are shown and also an interpolation between two images.

**Questions:**

Please provide more details and insight on the optimization implementation: how is the threshold used, what is the impact of the value of $T$, how do the obtained latents look like, and is it really easy to use a different sampler?

**Ethical Concerns:**

["NO or VERY MINOR ethics concerns only"]

**Final Justification:**

The rebuttal of the author did address my concerns. The influence of T and the threshold $\eta$ should be discussed in the main paper. Also, the optimal number of T steps: please include an explanation why some methods require a high value for optimal results (eg DDIM Inversion uses T=1000) and for some (including the proposed method) low?

**Limitations:**

Limitations are not discussed in the main paper.

**Quality:**

3

**Strengths And Weaknesses:**

### Strengths
* To my opinion the current manuscript is excellent in describing the setting of inverse diffusion, the difficulties and the proposed solution.

### Weakness
* The manuscript has some confusing elements, it puts emphasize on a 'learning framework' (*e.g.* in the abstract and introduction), and hence I was first convinced that $\epsilon_\theta$ would be **learned** to predict the inverse diffusion problem. But, after a second and third read, it seems that is not the case, this is a 'test-time algorithm', where the dynamic programming optimization is run for each example at test/evaluation time, without learning any parameters. In that sense nothing is learned. Please clarify that (eg in the introduction, below Eq 18, and in the experiments) and add pseudo code for the final algorithm.
* A part of the experimental setup is not clear to me, how is $T$ used. And why does smaller T values lead to better results? From the results it seems that using T=1 gives better results than using T=2 or T=4, while in general for Diffusion holds that a larger T value will provide better results.
* In the experiments there is a convergence threshold ($\eta=10^{-5}$), which is not discussed before. Please describe how the threshold is used during optimization.
* In **Bottom up computation** it is stated that *without loss of generality, we adopt the DDIM sampling strategy*. Would it be possible to use the DDPM strategy (Eq 6), and would that be easier or more difficult due to the additional noise term $z$?


### Remaining remarks
* I would like to get more insights in the optimization, how many steps (on average) are needed to reach the threshold? Is the resulting noisy image indeed a sample from a N(0,1) Gaussian? Is there a clear tradeoff between the used threshold, the number of steps T, and the image reconstruction quality.
* Table 2: How many iterations are used, ie what is the value for T?
* Table 2 & 3: It seems that using T=2 performs better than T=4 (on LPIPS, SSIM, PSNR). Is that expected? Is there a trade-off of quality versus number of steps T? (And is it better to have a few steps with a stricter convergence threshold or more steps with a tighter threshold?)
* In Figure 2, there are two $\epsilon^{*}_{t}$ depicted.
  * Fix: one of these should be $\epsilon^*_{t+1}$

---

> ### Author Rebuttal · Authors · 2025-07-31
>
> We sincerely thank the reviewer **SWAE** for their thoughtful feedback. The reviewer’s comments helped us identify important issues and improve the clarity and rigor of the paper.
> ***
> **Q1: The proposed method should be described as a test-time algorithm rather than a learning framework. The manuscript should clarify this distinction and include pseudocode for the final algorithm.**
>
> A1: Thank you for pointing out the misrepresentation, and we apologize for any confusion. The proposed method is indeed a test-time optimization algorithm, not a conventional learning framework. No model parameters are trained; instead, we perform gradient-based optimization for each test instance. We will revise the abstract, introduction, and conclusion to remove references to a "general learning framework" and clarify that our approach is a test-time procedure. Additionally, we will try to move the pseudocode for the DDIM-based version from Appendix B.2 to follow Eq. (18) in the main text.
>
> **Q2: In the experiments there is a convergence threshold $\eta$, which is not discussed before. Please describe how the threshold is used during optimization.**
>
> A2: We apologize for the lack of clarity. The convergence threshold $\eta$ sets an upper bound on the local reconstruction error for each subproblem $P(t)$ during optimization. Specifically, when the reconstruction error falls below $\eta$, we terminate the optimization for that subproblem and move on to the next. We will clarify this in both the method and experimental sections of the revised manuscript.
>
> **Q3: Table 2 & 3: Why does smaller T lead to better results?**
>
> A3: Great observation! As explained in A2, each subproblem is optimized until the local reconstruction error reaches the threshold $\eta$. When T increases, local errors accumulate, leading to higher global error and lower reconstruction quality. However, as shown in the table below, the error grows sublinearly with T, due to the Laplacian continuity of the diffusion latent space. From an efficiency standpoint, smaller T reduces the total number of optimization problems, speeding up inference. This underscores the core advantage of our method: accurate inversion with fewer steps, enabling both high-quality and fast real image reconstruction.
>
> |              | T=2    | T=5    | T=10   | T=20   | T=50   | T=100  |
> |--------------|--------|--------|--------|--------|--------|--------|
> | $\eta = 10^{-2}$ | 0.0025 | 0.0243 | 0.1363 | 0.1767 | 0.2015 | 0.2181 |
> | $\eta = 10^{-3}$ | 0.0019 | 0.0033 | 0.0091 | 0.0552 | 0.0721 | 0.1754 |
> | $\eta = 10^{-4}$ | **0.0019** | 0.0020 | 0.0020 | 0.0036 | 0.0431 | 0.0818 |
> | $\eta = 10^{-5}$ | -      | **0.0020** | **0.0019** | 0.0020 | 0.0033 | 0.0077 |
> | $\eta = 10^{-6}$ | -      | -      | -      | **0.0019** | 0.0020 | 0.0029 |
> | $\eta = 10^{-7}$ | -      | -      | -      | -      | **0.0019** | 0.0019 |
> | $\eta = 10^{-8}$ | -      | -      | -      | -      | -      | **0.0019** |
>
> **Q4:  Is there a trade-off of quality versus number of steps T?**
>
> A4: Thank you for the question. This trade-off is analyzed in Appendix C.2 (Lines 372–375). As shown in Table 5, with a fixed $\eta$, increasing T initially improves reconstruction quality and reduces inference time. However, beyond a certain point, both quality and efficiency decline. This reflects the trade-off: while larger T reduces initialization errors for each subproblem, it also increases the risk of error accumulation across steps.
>
> **Q5: In Bottom up computation it is stated that without loss of generality, we adopt the DDIM sampling strategy. Would it be possible to use the DDPM strategy (Eq 6), and would that be easier or more difficult due to the additional noise term?**
>
> A5: Thanks for the question! We have extended PreciseInv to support both the DDPM sampler and the Euler Discrete Sampler of flow matching. In the DDPM setting, the additional noise term in Eq. (6) is treated as a learnable variable and optimized during inference. Formal derivations for both extensions are provided in Appendix A.1, and we report quantitative results on a COCO 2017 subset in Table 4 of Appendix C.2. A direct comparison between DDIM and DDPM samplers using the SD v1.4 model is shown in the table below. The results demonstrate that our method converges reliably with both samplers and achieves comparable reconstruction quality. Interestingly, inference with DDPM is slightly faster, likely due to the added flexibility of the parameterized noise in fitting the generative trajectory. We will clarify this in the revised manuscript.
>
> | Sampler | LPIPS | SSIM  | PSNR  | Time (s) |
> |---------|-------|-------|-------|----------|
> | DDPM    | 0.080 | 0.764 | 25.93 | 9.08     |
> | DDIM    | 0.079 | 0.761 | 25.83 | 10.05    |
>
> **Q6: Table 2: What is the value for T?**
>
> A6: We apologize for the confusion. In Table 2, we used the optimal number of inference steps T for each method to ensure a fair comparison in reconstruction quality. As detailed in Appendix B.2: DDIM Inversion uses T=1000; ReNoise and BDIA use T=50; EasyInv and BELM use T=100; and both EDICT and our PreciseInv use T=2. We will clarify this setting in the main text in the revised version.
>
> **Q7: In Figure 2, there are two $x_t$ depicted. One of them should be $x_{t+1}$.**
>
> A7: Thank you for pointing this out. We will fix the notation error in Figure 2 in the revised version.
>
> **Q8: Please provide more details and insight on the optimization implementation: how is the threshold used, what is the impact of the value of $\eta$, how do the obtained latents look like, and is it really easy to use a different sampler?**
>
> A8: Sorry for the unclear. For details on the use and impact of the convergence threshold $\eta$, please refer to A2. In addition, our method is sampler-agnostic and can be easily adapted to DDPM and other samplers, as shown in A5.
>
> **Q9: I would like to get more insights in the optimization, how many steps (on average) are needed to reach the threshold? Is the resulting noisy image indeed a sample from a N(0,1) Gaussian? Is there a clear tradeoff between the used threshold, the number of steps T, and the image reconstruction quality.**
>
> A9: Thank you for the insightful question. We computed the KL divergence between the optimized $x_T$ from 100 randomly sampled test images and 100 samples from a standard Gaussian $N(0,1)$. The divergence was below $10^{−2}$, indicating that the optimized latents are effectively drawn from a Gaussian distribution. We will also include visualizations of the optimized latents in the appendix in our revised version. Please refer to **A4** for the discussion on the tradeoff between parameters and reconstruction outcomes.

---

> > ### Comment · Reviewer_SWAE · 2025-08-01
> > **Response to Rebuttal**
> >
> > Thank you for the detailed response on my main concerns, these are largely taken away now.
> > Please consider moving some essential information from the appendices to the main body of the paper, as this will improve the understanding of the method (not everyone reads and the paper and all the appendices!).

---

> > > ### Author Response · Authors · 2025-08-01
> > >
> > > Thank you for the encouraging feedback. We are glad that the main concerns have been addressed. We appreciate the suggestion and will incorporate key information from the appendix into the main text in the revised version to improve clarity and accessibility.

---

### Official Review · Reviewer_48AE · 2025-07-02

**Clarity:** 1
**Significance:** 2
**Originality:** 2
**Rating:** 3
**Confidence:** 3

**Summary:**

This paper proposes to reframe the diffusion inversion problem as a series of step-wise optimization sub-problems, rather than treating it as a traditional fixed-point problem. The method specifically addresses challenges related to accuracy and efficiency in inverting few-step inference diffusion models. Experimental results on both reconstruction and editing tasks demonstrate the effectiveness of the proposed method.

**Questions:**

- Q1. *(This question arose before I read Appendix B, but I'm still uncertain whether my interpretation is correct, so I'm leaving it here for clarification.)* What is the architecture of the PreciseInv model? Given that it appears to be comparable in size to the denoising backbone $\epsilon_\theta$, wouldn't such a large model introduce inefficiencies for inversion? Moreover, according to Eq. 20, $\epsilon_\theta$ must also reside in memory during PreciseInv training. Has the computational burden imposed by this requirement been analyzed, particularly in terms of limiting model size?
  - Training denoisers for T2I diffusion models is computationally expensive. Wouldn't training an inversion model that generalizes across diverse prompts and image (=$x_0$) also require substantial data and training time? How much training data is required?
  - Sec. 4.1 mentions only a validation set. What dataset was used for training?

- Q2. According to l29-32, backward optimization methods improve reconstruction quality by following the reverse diffusion trajectory. However, l179-180 claim these methods are not designed for accurate reconstruction and are excluded from comparison. Isn't this contradictory?
- Q3. What does the threshold $\eta$ in l186 represent?
- Q4. In Table 1 and Table 2, are the number of inversion steps for the proposed method and the baseline algorithms the same? The proposed method appears to use gradient descent to optimizer $\epsilon_t^*$, which involves both forward and backward passes through the diffusion model at each iteration. Since one gradient descent step is unlikely to suffice per time step, it seems that the proposed method would require more iterations to reach a final output if the number of time steps is fixed. Then, why is it claimed to be faster?
- Q5. In Sec. 3.3 on prompt-driven image editing, what is the rationale for denoising from the inverted latent $x_T^*$ using Eq. 21? Is there any theoretical justification for Eq. 21? Why is denoising not performing using $x_{c_t, \emptyset}$?
- Q6. In Figure 5, what prompt was used for denoising the interpolated latent $x_T^*$? Were interpolation-aware prompts also used in the baseline methods for a fair comparison?

**Ethical Concerns:**

["NO or VERY MINOR ethics concerns only"]

**Final Justification:**

During the author discussion phase, the authors addressed the concerns by clarifying the mis-framing and missing details of the proposed method. They also presented additional experiments, via PIE-Bench, demonstrating applications which inversion problem is fundamentally targeting. Furthermore, they provided theoretical justification for the proposed method to align with general NeurIPS standards. While the authors' responses to the reviewers' concerns necessitate substantial revisions to the original manuscript, the updated version still lacks the PIE-Bench experimental results and theoretical justification regarding convergence. This leaves some uncertainty about the final form of the paper. Accordingly, my final rating reflects both positive assessment of the resolved concerns and the remaining issues that have yet to be addressed.

**Limitations:**

The authors do not discuss the limitations of their proposed method

**Quality:**

2

**Strengths And Weaknesses:**

**Strengths**
- S1. The proposed inversion method appears reasonable, and the experimental results are convincing.
- S2. The proposed method performs well on few-step inference diffusion models such as LCM, showing improvements over comparison methods.

**Weaknesses**
- W1. The most critical issue is that, despite the paper framing the method as a **learning-based** inversion approach, it does not actually involve **learning** in the conventional sense. Throughout the main text, it is unclear what model or parameters are being trained. Only in Appendix B is it clarified that the proposed method **directly optimizes** the inverted noise, indicating that it is a direct optimization procedure rather than a learning-based model. This mischaracterization may mislead readers. Conventionally, learning-based inference using parametric or non-parametric models is expected to perform worse than direct optimization. Therefore, presenting the method as "learning-based" could result in an overestimation of its generality and performance.
- W2. The analysis and experimental results on image editing is insufficient. In the context of image generation models, the primary motivation for inversion is to enable controlled editing through manipulation of the inverted input or generation trajectory. However, the editing examples provided are limited to minor modifications. Color editing reflects only simple intensity changes, and style transfer is demonstrated only through sketching or pixelation, lacking complexity. Moreover, no quantitative evaluation is provided. The authors should include a broader range of edit prompts and define a standardized prompt set for benchmarking, to ensure reproducibility and diversity. Additionally, quantitative metrics and user studies are necessary to objectively validate editing performance.
- W3. The set of baseline algorithms is limited and includes outdated methods. To better establish the advantages of the proposed method, comparisons with more diverse and recent works are needed.

---

> ### Author Rebuttal · Authors · 2025-07-31
>
> We sincerely thank reviewer **48AE** for the critical and constructive feedback! Below, we provide point-by-point responses to the specific questions and concerns raised, and we hope these clarifications adequately address the issues.
> ***
> **Q1: The paper should avoid presenting the method as a learning-based inversion approach, as it primarily functions at inference time. In addition, the denoising backbone appears overly large, which may result in suboptimal efficiency for the inversion task.**
>
> A1: Thank you for pointing this out, and sorry for the confusion. Our method is indeed a test-time optimization approach, not a conventional learning-based method. We will revise the manuscript to avoid potentially misleading terms like "learning framework" and clarify that no offline training is involved.
>
> Regarding the backbone, we agree it may be overparameterized for simple reconstructions. However, we intentionally retain the original denoiser (e.g., SDXL) to ensure faithful alignment with the model’s generative process. This design choice reflects the flexibility of our framework, it can readily accommodate lighter or customized backbones if efficiency is prioritized.
>
> However, we also want to emphasize that our contributions remain despite the imprecise phrasing, i.e., (i) a general framework for diffusion inversion based on sequential subproblem solving; (ii) a dynamic programming-inspired strategy for efficient optimization; (iii) state-of-the-art reconstruction quality; and (iv) latent inference that preserves semantic continuity in the diffusion space.
>
> **Q2: The analysis and experimental results on image editing is insufficient.**
>
> A2: Thank you for the critical comments. While our method is designed as a general-purpose diffusion inversion framework (as noted in Lines 219–224) rather than a dedicated image editing algorithm, we agree that standardized evaluation on editing tasks is valuable.
>
> To address this, we now include quantitative results on the PIE-Bench subset (as suggested by reviewer **4z49**) using the SD v1.4 model. As shown in the table below, our method significantly outperforms baselines in structural consistency, background preservation, and CLIP score, consistent with the qualitative results in Figure 4.
>
> |               | Methods       | Structure Distance | Background Preservation |       |       |       | CLIP Score |
> |---------------|---------------|--------------------|--------|-------|-------|-------|------------|
> |               |               |                    | PSNR   | LPIPS | MSE   | SSIM  |            |
> | Invertible Samplers| EDICT         | 24.70              | 27.25  | 59.20 | 38.83 | 90.36 | 23.82      |
> |               | BDIA    | 35.91              | 22.29  | 78.27 | 58.95 | 87.26 | 24.04      |
> |               | BELM          | 11.25              | 28.55  | 51.44 | 33.93 | 90.44 | 24.25      |
> |               | DDIM Inversion| 72.28              | 17.61  |186.04 |173.05 | 80.74 | 23.61      |
> | Forward Optimizing | ReNoise  | 12.36              | 28.41  | 67.32 | 29.81 | 87.05 | 22.18      |
> |               | PreciseInv    | **9.44**           |**30.29**|**44.98**|**13.69**|**91.47**|**25.16**|
>
> **Q3: According to l29-32, backward optimization methods improve reconstruction quality by following the reverse diffusion trajectory. However, l179-180 claim these methods are not designed for accurate reconstruction and are excluded from comparison. Isn't this contradictory?**
>
> A3: Thank you for pointing this out. We apologize for the confusion. Lines 179–180 aim to clarify that backward optimization methods do not reconstruct the original generative trajectory of the diffusion model. Instead, they rely on recording the forward noising process and then manually adjusting the denoising steps to match the target image, thus effectively bypassing the need for optimization.
>
> While this enables accurate reconstructions in some cases, it deviates from the generative process and limits generalizability. Therefore, consistent with recent works focusing on forward optimization [a–d], we exclude these methods from direct comparison in our reconstruction experiments.
>
> [a] ReNoise: Real image inversion through iterative noising. ECCV, 2024.
>
> [b] EASYINV: Toward fast and better DDIM inversion. ArXiv, 2025.
>
> [c] Lightning-fast image inversion and editing for text-to-image diffusion models. ICLR, 2025.
>
> [d] Fixed-point inversion for text-to-image diffusion models. ArXiv, 2023.
>
> **Q4: What does the threshold $\eta$ in l186 represent?**
>
> A4: Apologies for the confusion. The threshold η defines the stopping criterion for each subproblem P(t): once the reconstruction error falls below η, optimization proceeds to the next step. It provides a flexible trade-off between accuracy and inference time. We will clarify this in the main text.
>
> **Q5: In Table 1 and Table 2, are the number of inversion steps for the proposed method and the baseline algorithms the same?**
>
> A5: Sorry for the unclear. As described in Appendix B.2 (Lines 333–339), we tuned the number of inference steps T and other hyperparameters for each baseline on 100 randomly sampled COCO images to ensure optimal reconstruction. The same settings were then applied across all datasets to avoid overfitting, ensuring a fair and reliable comparison.
>
> **Q6: Gradient-based optimization typically requires multiple iterations. Why, then, is it faster than numerical solvers?**
>
> A6: Good observation, thank you for the insightful feedback. As shown in Tables 1 and 2, our method achieves comparable reconstruction quality with significantly lower inference time. This efficiency comes from two key factors: (i) our approach uses far fewer diffusion steps (e.g., $T=2$) compared to the dozens or hundreds used by baseline methods; and (ii) gradient-based optimization leverages GPU acceleration through modern deep learning frameworks. To address your concern directly, we provide additional results showing that with $\eta=10^{-2}$ and $T=2$, the local error drops smoothly and meets the threshold within just 50 steps, requiring only 3.61 seconds in total. We will clarify this in the revised version.
>
> | Time (s) | 0.076 | 0.153 | 0.362 | 0.779 | 1.895 | 3.61 |
> |----------|-------|-------|-------|-------|-------|------|
> | Step     |   1   |   2   |   5   |  10   |  25   |  50  |
> | MSE      | 0.14  | 0.11  | 0.08  | 0.06  | 0.02  | 0.01 |
>
> **Q7: In Sec. 3.3 on prompt-driven image editing, what is the rationale for denoising from the inverted latent  using Eq. 21? Is there any theoretical justification for Eq. 21? Why is denoising not performing using ?**
>
> A7: Thank you for the thoughtful question. Following prior work [a], we adopt the standard inversion-then-generation paradigm without altering the editing pipeline. In Eq. (21), we reuse the inverted latent $x_T^*$, swapping the prompt roles—using the target as the positive prompt and the source as the negative, to steer generation away from the original semantics. This heuristic is widely used in existing methods. We also tested removing classifier-free guidance (CFG) during inversion (Eq. 19) and observed similar results, suggesting that CFG primarily influences sampling rather than inversion.
>
> [a] ReNoise: Real image inversion through iterative noising. ECCV, 2024.
>
> **Q8: In Figure 5, what prompt was used for denoising the interpolated latent ? Were interpolation-aware prompts also used in the baseline methods for a fair comparison?**
>
> A8: Thank you for the question. In Figure 5, we used an empty string as the prompt, and all methods used the same prompt to ensure a fair comparison. To further validate this, we also tested with a category prompt like “a photo of a car” and observed consistent results, i.e., our method produces an $x_T$ that lies in a smoother semantic region of the latent space. Prior work [e] shows that diffusion latent spaces are semantically continuous. By accurately recovering the model’s native generative trajectory, our approach yields latent representations that better align with this structure.
>
> [e] Unsupervised representation learning from pre-trained diffusion probabilistic models. NeurIPS, 2022.

---

> ### Comment · Reviewer_48AE · 2025-08-06
>
> I appreciate the author's detailed response. The rebuttal helped me clearly understand that the proposed method follows an optimization-based inversion approach. I believe that inversion is inherently tied to tasks such as image editing and manipulation. If the sole objective were to reconstruct a given image, there would be no need to go through the inversion and re-generation process, since the image is already available. From this perspective, I find it valuable contribution that the authors validated the effectiveness and superiority of their method on PIE-Bench, as suggested by Reviewer 4z49. Although the response could not include visual examples due to the policy constraints, it would be beneficial to include representative PIE-Bench examples in the manuscript.
>
> Aside from the initial mischaracterization as a learning-based approach, I do recognize originality in the proposed method. However, as pointed by Reviewer 4z49, the absence of formal theoretical guarantees, such as a convergence proof or error bounds, seems to remain a significant limitation. Including such mathematical analysis could help the technical soundness of the work and align it more closely with the standards typically expected at NeurIPS.

---

> > ### Author Response · Authors · 2025-08-07
> >
> > Dear reviewer 48AE,
> >
> > Thank you for your thoughtful and insightful comments. We sincerely appreciate your recognition of the originality of our work and your positive remarks on the PIE-Bench results. As suggested, we will include PIE-Bench visualizations in the revised version.
> >
> > Regarding your concern about theoretical guarantees, we have provided a more formal analysis of the convergence behavior and reconstruction error bounds in our response to Reviewer 4z49 (see above). We hope this additional perspective helps address the issue and strengthens the theoretical soundness of our method.

---

### Official Review · Reviewer_v5Ud · 2025-07-03

**Clarity:** 3
**Significance:** 3
**Originality:** 3
**Rating:** 4
**Confidence:** 3

**Summary:**

The paper proposes a diffusion inversion method that allows few-step inversion and generalization to novel samples by reformulating diffusion inversion as a progressive learning problem and using DP like strategy to solve this. The proposed method outperforms baseline methods in most settings and the difference is the most pronounced in the few-step inference setting.

**Questions:**

- Can the method be used with shortcut models as well?
- Do you have a particular explanation as to why the proposed method results in a smoother interpolation between latents? It would be interesting to know a more detailed reasoning behind this.
- It is unclear as to whether the method is viable for SD3 just looking at Table 3 as there is no baseline comparison. How does this inference speed compare to other baselines in SD3?

**Ethical Concerns:**

["NO or VERY MINOR ethics concerns only"]

**Final Justification:**

The provided method shows promise especially regarding the qualitative demonstration and out-of-distribution examples such as flat style illustrations. The authors addressed most of my concerns through their rebuttal, including application to more recent paradigm (flow matching DiTs). Therefore, I think this work is worth being considered for NeurIPS conference, and I keep my score of 4.

**Limitations:**

yes

**Quality:**

3

**Strengths And Weaknesses:**

**Strengths**
- By solving subproblems separately, the given method is more memory-efficient compared to baseline methods, which might be helpful for applying to larger models.
- The proposed method consistently shows better reconstruction accuracy compared to baseline methods with a much faster speed (Table 1).
- The proposed method works better than baseline methods especially under a few-step setting.
- Qualitative examples are still impressive, including the editing applications.
- Performing slerp between inverted latents results in a better qualitative result across all interpolated positions.
- The method can be extended to rectified flow models such as SD3 as well (Appendix A.2), and still shows a good reconstruction performance..

**Weaknesses**
- When adjusting the convergence threshold, the computation time seems to increase quite a lot, especially in the the case of SDXL. The speed-quality tradeoff needs to be discussed.
- In fact, the method seems to be quite slower for SD3 (Appendix C.1, Table 4), which may suggest that the computation time saving might not scale to larger models such as SD3 or FLUX, which can limit the applicability. A comparison against baseline methods for SD3 or FLUX might also be necessary.

---

> ### Author Rebuttal · Authors · 2025-07-31
>
> We sincerely thank reviewer **v5Ud** for the insightful feedback! The suggestions have helped us further clarify key aspects of the work.
> ***
> **Q1: When adjusting the convergence threshold, the computation time seems to increase quite a lot, especially in the case of SDXL. The speed-quality tradeoff needs to be discussed.**
>
> A1: We thank the reviewer for the helpful suggestion. We, indeed, have analyzed the speed–quality tradeoff under different convergence thresholds $\eta$ in Table 5 (Appendix C.1). Empirically, smaller $\eta$ leads to better reconstruction quality (lower LPIPS, higher SSIM/PSNR) but increases inference time due to stricter convergence.
>
> We also observe that this effect is more pronounced for large-scale models such as SDXL, as noted by the reviewer. This is expected, since SDXL has a deeper denoising UNet and a larger latent space, which increases the per-iteration computation cost. However, PreciseInv provides a flexible tradeoff mechanism: users can adjust $\eta$ based on available computational budget. For example, using a relatively loose threshold (e.g., $\eta = 10^{-2}$ with T=2), PreciseInv still achieves competitive reconstruction performance on SDXL while being significantly faster than optimization-based baselines.
>
> **Q2: The method seems to be quite slower for SD3, which may suggest that the computation time saving might not scale to larger models such as SD3 or FLUX, which can limit the applicability. A comparison against baseline methods for SD3 or FLUX might also be necessary.**
>
> A2: Thank you for pointing this out. We acknowledge that inference time on SD3 is longer than on SDv1.4 or SDXL due to its larger model size. However, accurately inverting the generative trajectory of such a large-scale model is inherently challenging, and to the best of our knowledge, our method is the first to achieve this effectively.
>
> Since our method belongs to the forward optimization family, it is generally not compared with backward correction-based approaches like [a, b]. While method [c] also follows a forward approach, its implementation is only available for SDXL-Turbo, not SD3 or Flux.
>
> Thus, we compare against FPI on SD3 with carefully tuned settings. As shown in the following table, although FPI is faster, our method significantly outperforms it in LPIPS, SSIM, and PSNR. Qualitative results further reveal that FPI reconstructions are often blurry or semantically inconsistent. We will include these results and discussions in the revised version.
>
> |                | LPIPS  | SSIM  | PSNR  | Time    |
> | -------------- | ------ | ----- | ----- | ------- |
> | Fixed Point Inversion | 0.6080 | 0.3121 | 13.01 | 226.73  |
> | PreciseInv (Ours)     | 0.104  | 0.896 | 30.12 | 403.31  |
>
> [a] Stable Flow: Vital Layers for Training-Free Image Editing. CVPR, 2025.
>
> [b] Unveil Inversion and Invariance in Flow Transformer for Versatile Image Editing. CVPR, 2025.
>
> [c] Lightning-fast image inversion and editing for text-to-image diffusion models. ICLR, 2025.
>
> **Q3: Can the method be used with shortcut models as well?**
>
> A3: Good spot! Yes, PreciseInv can be naturally extended to Shortcut models [d], as both SD3 and Shortcut adopt the flow-matching paradigm. Our method does not rely on any specific architecture; it only requires a differentiable and continuous mapping from $\mathbf{x} _ t$ to $\mathbf{x}_{t-\Delta}$, which remains valid in both SD3 and Shortcut. Although Shortcut introduces a step-size indicator $\Delta$ and reduces parameter count, it preserves key properties such as Lipschitz continuity of the velocity field. Since PreciseInv is already formally defined and empirically validated on SD3, it can be directly applied to Shortcut without modification. We plan to support this extension in our public code release.
>
> [d] One Step Diffusion via Shortcut Models. ICLR, 2025.
>
> **Q4: It would be interesting to know a more detailed reasoning behind why the proposed method results in a smoother interpolation between latents.**
>
> Thank you for the insightful question. Prior work [e] has demonstrated that the latent space of diffusion models exhibits inherent semantic continuity. Our method does not introduce any explicit regularization or interpolation constraints. Instead, it accurately reconstructs the model’s own generative trajectory, allowing the inferred latent $\mathbf{x} _ T$to reside naturally within a semantically coherent region. This results in significantly smoother interpolation between latents.
>
> We further conducted empirical analysis and observed that this continuity deteriorates as the reconstruction precision decreases. In particular, when the convergence threshold $\eta$ exceeds approximately 0.025, the interpolated results begin to show noticeable semantic discontinuities. This observation underscores the importance of precise inversion in preserving the smoothness and structure of the semantic latent space. However, the deeper mechanism behind this phenomenon remains an open question and is an exciting direction for future exploration.
>
> [e] Unsupervised representation learning from pre-trained diffusion probabilistic models. NeurIPS, 2022.

---

> > ### Comment · Reviewer_v5Ud · 2025-08-09
> >
> > I appreciate the authors for their detailed response, and especially the table for SD3 is reassuring, as this much computational cost overhead seems justifiable for this level of gain. I think this could be a useful tool in practice. Therefore, I keep my score of 4.

---

### Official Review · Reviewer_4z49 · 2025-07-03

**Clarity:** 2
**Significance:** 3
**Originality:** 3
**Rating:** 4
**Confidence:** 4

**Summary:**

The paper proposes PreciseInv, which is a novel learning-based method for diffusion model inversion that achieves accurate and efficient inversion in a few inference steps. The previous methods are usually based on fixed-point or root-finding. Different from them, PreciseInv reformulates inversion as a progressive learning task. In detail, it leverages the Markov property of diffusion models. It decomposes the problem into several overlapping subproblems and solves them using a dynamic programming-inspired strategy. The authors conduct experiments on COCO 2017, DarkFace, and a cartoon dataset. The results show that PreciseInv achieves better reconstruction quality and efficiency compared with existing methods. Besides, it also improves semantic consistency in image editing.

**Questions:**

1. Could the author provide the analysis of convergence or approximation error about the optimization?
2. Could the author provide a quantitative comparison in image editing? BIE-Benchmark is a good choice.
3. How sensitive is the method to the choice of threshold?

**Ethical Concerns:**

["NO or VERY MINOR ethics concerns only"]

**Final Justification:**

The authors have addressed my concerns regarding the image editing evaluation and the choice of convergence threshold. They also provided additional convergence analysis. However, formal theoretical guarantees are still lacking. Therefore, I will maintain my original score.

**Limitations:**

Yes

**Quality:**

3

**Strengths And Weaknesses:**

Strengths:
1. The paper addresses an important problem. Diffusion inversion is important for image editing. The current fixed-point and root-finding method cannot find an accurate noise embedding. The paper aims to propose an accurate and efficient inversion method.
2. The method is novel. The paper reformulates diffusion inversion as a learning problem. It also decomposes the problem into recursive subproblems and proposes a dynamic programming-inspired strategy to solve it.
3. The results are good. The proposed method achieves strong trade-offs between reconstruction accuracy and efficiency under a step inference. The method also improves semantic consistency and controllability in image editing. The results demonstrate good performance in various datasets and diffusion architecture.


Weaknesses:
1. The theoretical analysis is missing. Although the proposed method is straightforward, the analysis of convergence or approximation error in the optimization should be provided.
2. Comparison in image editing is limited. The paper only provides a qualitative comparison of image editing. However, quantitative metrics are missing.
3. Some ablation studies are missing. The paper uses different thresholds. The studies about the choice should be provided.

---

> ### Author Rebuttal · Authors · 2025-07-31
>
> Our sincere thanks to Reviewer **4z49** for their thoughtful and insightful comments. We've addressed each of the specific questions and concerns they brought up in our detailed, point-by-point responses below.
> ***
> **Q1: The theoretical analysis is missing. Although the proposed method is straightforward, the analysis of convergence or approximation error in the optimization should be provided.**
>
> A1: Thank you for the suggestion. While our method is primarily empirical, we provide convergence-related evidence through additional experiments shown in Table 1 and Table 2. Table 1 illustrates the local reconstruction error across iterations for each subproblem $P(t)$, showing smooth convergence. This behavior stems from the Lipschitz continuity of the diffusion latent space, which facilitates stable gradient-based optimization. Table 2 analyzes the global reconstruction error over T subproblems under varying upper bounds $\eta$ on local error. Notably, global error does not accumulate linearly with T. Moreover, fixing T and progressively reducing $\eta$ leads to a converged global error around 0.0019, which reflects the intrinsic reconstruction limit of the VAE in latent diffusion. These findings demonstrate that our method enjoys stable convergence and favorable approximation accuracy under varying inference steps T. We will incorporate these results and discussions into our revised version.
>
> **Table 1: Local reconstruction error at each optimization step in subproblem $P(t)$. Results are averaged over 32 images from the COCO 2017 validation set.**
> | Step | 1    | 2    | 5    | 10   | 25   | 50   |
> |------|------|------|------|------|------|------|
> | $\|\|\mu(x_t^*)-x_{t-1}^\*\|\|^2$ | 0.14 | 0.11 | 0.08 | 0.06 | 0.02 | 0.01 |
>
> **Table 2: Global accumulated reconstruction error over $T$ subproblems. The hyperparameter $\eta$ defines the error bound for each subproblem $P(t)$: if and only if the local reconstruction error drops below $\eta$, the optimization moves to the next subproblem $P(t+1)$.**
> |                | T=2           | T=5           | T=10          | T=20          | T=50          | T=100         |
> | -------------- | ------------- | ------------- | ------------- | ------------- | ------------- | ------------- |
> | $\eta = 10^{-2}$ | 0.0025        | 0.0243        | 0.1363        | 0.1767        | 0.2015        | 0.2181        |
> | $\eta = 10^{-3}$ | 0.0019 | 0.0033        | 0.0091        | 0.0552        | 0.0721        | 0.1754        |
> | $\eta = 10^{-4}$ | **0.0019**    | 0.0020 |0.0020 | 0.0036        | 0.0431        | 0.0818        |
> | $\eta = 10^{-5}$ | -             | **0.0020**    | **0.0019**    |0.0020 | 0.0033        | 0.0077        |
> | $\eta = 10^{-6}$ | -             | -             | -             | **0.0019**    | 0.0020 | 0.0029        |
> | $\eta = 10^{-7}$ | -             | -             | -             | -             | **0.0019**    | 0.0019 |
> | $\eta = 10^{-8}$ | -             | -             | -             | -             | -             | **0.0019**    |
>
> **Q2: Could the author provide a quantitative comparison in image editing? PIE-Benchmark is a good choice.**
>
> A2: Thank you for the construction comments. Following your suggestion, we provide additional quantitative results on PIE-Bench [a] in Table 3, where PreciseInv significantly outperforms baseline methods in terms of structural consistency, background preservation, and instruction fidelity. This improvement stems from our accurate reconstruction of the original generative trajectory, which retains more information from the source image $\mathbf{x}_0$ and produces a latent $\mathbf{x}_T$ situated in a smoother, semantically meaningful space. These quantitative results are also consistent with the qualitative comparisons shown in Fig. 4.
>
> [a] PNP Inversion: Boosting diffusion-based editing with 3 lines of code. ICLR, 2024.
>
> **Table 3: Quantitative results on the PIE-Bech subset. For each category, we randomly sample 10 test instances. We adopt Stable Diffusion v1.4 as the base model for all methods.**
>
> |               | Methods       | Structure Distance | Background Preservation |       |       |       | CLIP Score |
> |---------------|---------------|--------------------|--------|-------|-------|-------|------------|
> |               |               |                    | PSNR   | LPIPS | MSE   | SSIM  |            |
> | Invertible Samplers| EDICT         | 24.70              | 27.25  | 59.20 | 38.83 | 90.36 | 23.82      |
> |               | BDIA    | 35.91              | 22.29  | 78.27 | 58.95 | 87.26 | 24.04      |
> |               | BELM          | 11.25              | 28.55  | 51.44 | 33.93 | 90.44 | 24.25      |
> |               | DDIM Inversion| 72.28              | 17.61  |186.04 |173.05 | 80.74 | 23.61      |
> | Forward Optimizing | ReNoise  | 12.36              | 28.41  | 67.32 | 29.81 | 87.05 | 22.18      |
> |               | PreciseInv    | **9.44**           |**30.29**|**44.98**|**13.69**|**91.47**|**25.16**|
>
> **Q3: How sensitive is the method to the choice of the convergence threshold $\eta$?**
>
> A3: Thank you for the question. In fact, we have thoroughly analyzed the sensitivity of our method to the convergence threshold $\eta$, as shown in Table 5 of Appendix C.3 in the original manuscript.
> We observe that lowering $\eta$ from $10^{-2}$ to $10^{-5}$ consistently improves reconstruction quality across metrics such as LPIPS, SSIM, and PSNR. However, further tightening (e.g., to $10^{-6}$ or $10^{-7}$) leads to diminishing returns, especially when the number of inference steps T is small.
>
> In terms of efficiency, smaller $\eta$ values do increase inference time. Nevertheless, thanks to the smoothness of the latent space and our progressive optimization strategy, the runtime remains practical (e.g., 5.33s per image at $\eta = 10^{-5}$) and becomes very fast under relaxed settings like $\eta = 10^{-2}$.
>
> Crucially, PreciseInv remains robust across a wide range of $\eta$ values, outperforming all baselines in both accuracy and speed even with loose thresholds. This indicates that the method does not require fine-grained tuning to deliver strong and reliable results.

---

> > ### Comment · Reviewer_4z49 · 2025-08-06
> >
> > Thanks for the detailed responses. The authors provide convergence analysis and PIE-Benchmark results, which are very helpful. However, the lack of formal theoretical guarantees, such as convergence proof or error bound, is still a limitation. Therefore, I decide to maintain my original score.

---

> > > ### Author Response · Authors · 2025-08-06
> > >
> > > Thank you for your initial positive rating and constructive comments. We appreciate your time and will continue working to strengthen the theoretical aspects in future work.

---

> ### Author Response · Authors · 2025-08-07
> **Formal theoretical proof**
>
> Thank you for your constructive comments. After a deeper investigation of this issue, we have now developed a more **formal** framework for analyzing convergence behavior and bounding reconstruction error. We hope this provides a clearer theoretical perspective and better addresses your concern.
>
> We will include the formal proof outlined below in our revised version.
> ***
>
> **Problem Setup**
>
> We formulate diffusion inversion as an optimization problem over a parameterized noise sequence $\\{\epsilon_ t\\}_{t=1}^T$, aiming to minimize the reconstruction error:
>
> $\min _ {\\{\epsilon_t\\}} \mathcal{L} := \||x_0 - \hat{x} _ 0(\epsilon_{1:T})\||^2$
>
> The image reconstruction $\hat{x}_0$ is obtained by applying the backward denoising steps of a pre-trained diffusion model:
>
> $x_{t-1} = f_\theta(x_t, \epsilon_t), \quad t = T, \ldots, 1$
>
> We further decompose this into local subproblems using a dynamic programming-style recursive strategy, optimizing $\epsilon_t$ sequentially to reduce local errors:
>
> $\mathcal{L} _ t := \||x _ {t-1} - f_\theta (x_t, \epsilon_t)\||^2$
>
> **Assumptions**
>
> To enable convergence and error analysis, we introduce the following assumptions:
> - *(A1) Lipschitz Continuity of the Model*:
> The denoising function $f_\theta(x_t, \epsilon_t)$ is Lipschitz continuous in $\epsilon_t$ with constant $L_m > 0$, i.e.,
> $$ \||f_\theta(x_t, \epsilon) - f_\theta(x_t, \epsilon')\|| \leq L_m \cdot \||\epsilon - \epsilon'\|| \quad \forall \epsilon, \epsilon'$$
> - *(A2) Lipschitz Smoothness of the Loss Function*:
> The local reconstruction loss $\mathcal{L}_t(\epsilon_t)$ has $L$-Lipschitz continuous gradients with respect to $\epsilon_t$, i.e.,
> $$\||\nabla \mathcal{L}_t(\epsilon) - \nabla \mathcal{L}_t(\epsilon')\|| \leq L \cdot \||\epsilon - \epsilon'\|| \quad \forall \epsilon, \epsilon'$$
> This constant $L$ governs the smoothness of the loss landscape and is crucial for step-size selection in gradient-based optimization.
> - *(A3) Locally Bounded Gradients*:
> Within a compact neighborhood $\mathcal{B}_r(\epsilon_t^{(0)})$ around the initialization of $\epsilon_t$, the gradient is uniformly bounded:
> $$\||\nabla \mathcal{L}_t(\epsilon_t)\|| \leq G, \quad \forall \epsilon_t \in \mathcal{B}_r(\epsilon_t^{(0)})$$
> This assumption reflects empirical behavior in diffusion models, where network normalization (i.e., denoising backbones UNet or Transformer commonly use BatchNorm, GroupNorm or LayerNorm, which control the scale of intermediate activations), bounded input noise (i.e., inversion typically starts with $\epsilon_t \sim \mathcal{N}(0, I)$, and optimization trajectories remain within a moderate neighborhood), and the smoothness of the latent space (i.e., diffusion latent space exhibits Lipschitz continuity) ensure stable gradients during inference. These properties collectively ensure that gradient norms do not explode and remain bounded throughout optimization.
>
> **Convergence Analysis**
>
> Under the above assumptions (i.e., each denoising function $f_\theta$ is Lipschitz continuous in $\epsilon_t$ with constant $L_t$, and the loss functions $\mathcal{L}_t$ are differentiable with bounded gradients), standard results from non-convex optimization apply [a]. Specifically, using gradient descent with a fixed learning rate $\eta < \frac{2}{L}$, we can obtain:
> - Each local optimization subproblem decreases its objective value monotonically.
> - After $K$ gradient steps, the minimum gradient norm satisfies:
> $$ \min_{k \in [1, K]} \||\nabla \mathcal{L}_t^{(k)}\||^2 \leq \mathcal{O}\left(\frac{1}{K}\right)$$
> indicating convergence to a stationary point.
> - Since each step optimizes a local reconstruction loss, and the overall optimization proceeds recursively, the global loss $\mathcal{L}$ is non-increasing across steps.
>
> **Error Bound Analysis**
>
> Let the estimation error at each step satisfy $\||\epsilon_t - \hat{\epsilon} _ t\|| \leq \delta_t$, and assume the mapping $x_{t-1} = f_\theta(x_t, \epsilon_t)$ is Lipschitz in $\epsilon_t$. Then the accumulated reconstruction error satisfies (following A1):
> $$\||x _ 0 - \hat{x}_0\|| \leq \sum _ {t=1}^T \left( \prod _ {j=1}^{t-1} L_m^{(j)} \right) \cdot \delta_t$$
> If $L_m^{(j)} \leq L_m < 1$ (satisfied due to Lipschitz continuity in diffusion latent space, and empirical observations in Table 1 in our initial response) and $\delta_t \leq \delta$ for all $t$, we obtain a geometric upper bound:
>
> $$\||x_0 - \hat{x}_0\|| \leq \delta \cdot \sum _ {t=1}^T L_m^{t-1} \leq \frac{\delta}{1 - L_m}$$
>
> This aligns with our empirical results (see our initial response Table 2), where the reconstruction error saturates and does not grow linearly with $T$.
>
> **Reference**
>
> [a] Stochastic first-and zeroth-order methods for nonconvex stochastic programming. SIAM journal on optimization 2013.

---

> ### Author Response · Authors · 2025-08-07
> **Formal theoretical proof - part 2**
>
> **Detailed proof of the geometric upper bound**:
>
> Assume that at each subproblem $t$, the local reconstruction error is bounded by $\delta_t \leq \delta$. Furthermore, suppose the error propagation factor at each stage is Lipschitz continuous with constant $L_m \in (0, 1)$, i.e.,
> $$\||\text{error at step } t\|| \leq L_m \cdot \||\text{error at step } t-1\|| + \delta$$
> Then, the global error accumulates as follows:
>
> $e_1 = \delta$,
>
> $e_2 \leq L_m \cdot e_1 + \delta = L_m \cdot \delta + \delta$
>
> $e_3 \leq L_m \cdot e_2 + \delta = L_m^2 \cdot \delta + L_m \cdot \delta + \delta$
>
> ...
>
> $e_T \leq \delta \cdot \sum_{k=0}^{T-1} L_m^k = \delta \cdot \frac{1 - L_m^T}{1 - L_m} \leq \frac{\delta}{1 - L_m}$
>
> Hence, the final error between the estimated and true initial latent is bounded by
> $$\||x_0 - \hat{x}_0\|| \leq \||e_T\|| \leq \frac{\delta}{1 - L_m}$$

---

> > ### Comment · Reviewer_4z49 · 2025-08-08
> >
> > Thanks for the update. I have a few questions:
> > 1. The A3 assumes $\| \nabla \mathcal{L}_t(\epsilon_t) \| \leq G$ within a local neighborhood. Do the optimization trajectories remain within this neighborhood throughout inference?
> > 1. Each $\epsilon_t$ is optimized locally to reduce $\mathcal{L}_t$. How does it guarantee that the overall reconstruction loss $\mathcal{L} = \|x_0 - \hat{x}_0\|^2$ is minimized?
> > 2.  The bound $\|x_0 - \hat{x}_0\| \leq \delta / (1 - L_m)$ assumes uniform error $\delta_t \leq \delta$ and constant $L_m$. Are these values empirically estimated? Are they consistent across time steps?

---

> ### Author Response · Authors · 2025-08-08
> **Clarifications of the proof**
>
> Thank you for your prompt and thoughtful questions. We are glad to continue this exchange and have provided detailed responses to each point below. We look forward to your thoughts on our clarifications.
> ***
> **1. Do the optimization trajectories remain within this neighborhood throughout inference?**
>
> **Re1**: Yes, the iterates do remain close to their initialization in our setting. Specifically:
>
>  - We initialize $\epsilon_t$ from the model’s typical starting guess ( $\epsilon_t \sim \mathcal{N}(0, I)$ ), so the initialization $\epsilon_t^{(0)}$ is already in a region the network was trained on (samples from $\mathcal{N}(0, I)$  or close variants).
>  - We perform only a small number of gradient steps per subproblem. Empirically, this yields minor changes per iteration, so the iterates stay within a moderate neighborhood of $\epsilon_t^{(0)}$. This follows naturally from the *Lipschitz continuity of the model* (Assumption A1).
>  - By setting the step size (learning rate) $\eta'$ to satisfy $\eta' < \frac{2}{L}$ (from the cited work [a]), we can further guarantee that the optimization trajectories remain within this neighborhood.
>
> **2. Each $\epsilon_t$ is optimized locally to reduce $\mathcal{L}_t$. How does it guarantee that the overall reconstruction loss $\mathcal{L}$  is minimized?**
>
> **Re 2**: According to the **convergence analysis** in our proof above, if $f_\theta$ is Lipschitz in $\epsilon_t$ (A1), then a local decrease in $\mathcal{L}_t$ reduces the downstream discrepancy $\\|x_0 - \hat{x}_0\\|$ by at most a Lipschitz factor. Formally, for a small local improvement $\Delta_t$, we have
> $$\\|x_0 - \hat{x}_0\\| \leq C_t \cdot \Delta_t$$
> where $C_t$ depends on the product of downstream Lipschitz constants $L_m^{(j)}$ ($L_m^{(j)} \leq L_m < 1$ due to Lipschitz continuity in the diffusion latent space). Thus, local descent yields **bounded global descent**:
> $$\\|x_0 - \hat{x} _ 0\\| \leq \delta \cdot \sum _ {t=1}^T L_m^{t-1} \leq \frac{\delta}{1 - L_m}$$
> Please refer to our **Formal Theoretical Proof - part 2** for more details on this **geometric upper bound**. Moreover, empirical observations suggest that the overall $\mathcal{L}$ decreases monotonically as subproblems are solved (see Table 2 in our response to Q1).
>
> **3. Are these values ($\delta_t$,$L_m$) empirically estimated? Are they consistent across time steps?**
>
> **Re 3**: Yes. These constants can be empirically estimated and, if desired, allowed to vary with timesteps $t$. For convenience, we often adopt conservative uniform bounds (i.e., consistent values across time steps). Specifically, we can
>
>   - Estimate $\delta_t$ (local error) by running the local optimization on a validation set and record the reconstruction error $\Delta_t = \\|x_t - \hat{x}_t \\|$ for each $t$. A uniform bound can then be set as $\delta = \max_t \Delta_t$.
>
>  - Estimate $L_m^{(j)}$ (local Lipschitz constant) by approximation via finite differences on a validation set:
> $$L_m^{(j)} \approx \max_{i} \frac{\left\\| f_{\theta}\left(x_t^{(i)}, \, \epsilon^{(i)} + \Delta\right) - f_{\theta}\left(x_t^{(i)}, \, \epsilon^{(i)} \right) \right\\|} {\\|\Delta\\|}$$
> for small random $\Delta$. A conservative bound is then $L_m = \max_j L_m^{(j)}$.
>
> In practice, we enforce a per-step accuracy threshold $\eta$ during subproblem optimization, i.e., we stop optimizing step $t$ once:
> $$\Delta_t := \\|x_t - \hat{x}_t\\| \le \eta$$
> This simple rule yields several beneficial effects:
>
>  -  Setting $\delta := \eta$ provides a uniform per-step upper bound, which directly yields the global bound
> $$\\|x_0 - \hat{x}_0\\| \le \delta \sum _ {t=1}^T L_m^{t-1} \le \frac{\delta}{1-L_m}$$ under standard Lipschitz assumptions.
>
>  - Thresholding empirically keeps iterates in a small neighborhood of the initialization, making the bounded-gradient assumption more realistic.
>
>  - As noted in our Q1 response in the initial rebuttal, $\eta$ introduces an explicit accuracy–efficiency trade-off: smaller $\eta$ improves final reconstruction at higher per-step cost, while larger $\eta$ speeds up inversion with a looser bound.

---

### Decision · Program_Chairs · 2025-09-17

**Decision:**

Accept (poster)

**Comment:**

This paper introduces a precise diffusion inversion framework that improves reconstruction fidelity and enables few-step generation. The problem is timely, and the approach is well-motivated and clearly presented. Reviewers appreciated the strong experimental results and the careful analysis. Concerns about incremental novelty, computational cost, and limited evaluation breadth were raised initially, but the rebuttal provided clarifications on efficiency, scalability, and the positioning of contributions relative to prior work. These responses helped mitigate the main reservations. Overall, the work is technically sound, addresses an important problem, and offers meaningful improvements over existing inversion methods.

I urge the Authors to incorporate all the promised changes in the final draft including the new experiments and convergence proofs and related discussions.